microsystems

acoustic focusing, glass microfluidics, acoustofluidics, thin glass

**Author for correspondence:**
Yo Tanaka
e-mail: yo.tanaka@riken.jp

# Enhancement in acoustic focusing of micro and nanoparticles by thinning a microfluidic device

Nobutoshi Ota[1], Yaxiaer Yalikun[1,2], Tomoyuki Suzuki[2], Sang Wook Lee[3], Yoichiroh Hosokawa[2], Keisuke Goda[3,4] and Yo Tanaka[1]

[1]Center for Biosystems Dynamics Research (BDR), RIKEN, 1-3 Yamadaoka, Suita, Osaka 565-0871, Japan
[2]Division of Materials Science, Nara Institute of Science and Technology, Ikoma, Takayama, Nara 630-0192, Japan
[3]Department of Chemistry, School of Science, The University of Tokyo, 7-3-1 Hongo, Bunkyo-ku, Tokyo 113-0033, Japan
[4]Japan Science and Technology Agency, 4-1-8, Honcho, Kawaguchi, Saitama 332-0012, Japan

NO, 0000-0001-5365-2211; YT, 0000-0003-1034-5718

The manipulation of micro/nanoparticles has become increasingly important in biological and industrial fields. As a non-contact method for particle manipulation, acoustic focusing has been applied in sorting, enrichment and analysis of particles with microfluidic devices. Although the frequency and amplitude of acoustic waves and the dimensions of microchannels have been recognized as important parameters for acoustic focusing, the thickness of microfluidic devices has not been considered so far. Here, we report that thin glass microfluidic devices enhance acoustic focusing of micro/nanoparticles. It was found that the thickness of a microfluidic device strongly influences its ability to focus particles via acoustic radiation, because the energy propagation of acoustic waves is affected by the total mass of the device. Acoustic focusing of submicrometre polystyrene beads and *Escherichia coli* as well as enrichment of polystyrene beads were achieved in glass microfluidic devices as thin as 0.4 mm. Modifying the thickness of a microfluidic device can thus serve as a critical parameter for acoustic focusing when conventional parameters to achieve this effect are kept unchanged. Thus, our findings enable new approaches to the design of novel microfluidic devices.

# 1. Introduction

Acoustic focusing has gained attention as a non-contact method for manipulating micro/nanoparticles found in biological [1,2] and industrial [3,4] fields. Particle focusing, including acoustic focusing, has been widely studied by using inertial force [5] as well as electric [6], magnetic [7] and optical [8] methods. These focusing techniques are highly compatible with quickly advancing technology of miniaturized instruments, e.g. microflow cytometry [9,10], that are suitable for manipulating small particles like cells [11]. Acoustic focusing, as a particle focusing technique, has also been applied to sorting [12–14], enrichment [15,16] and analysis [17,18] of particles. For example, acoustic focusing is employed in flow cytometry [19,20], where focused particles need to be precisely aligned in a microchannel to obtain high-quality images in continuous flow. In addition, some studies have reported that the force induced by acoustic waves does not affect cellular viability in microfluidic devices [21,22]. By virtue of these advantages, acoustic focusing is considered as an important technique for investigating particles in the confined space within microfluidic devices.

To achieve a high degree of acoustic focusing, some important parameters need to be considered: the frequency and amplitude of acoustic waves, the design of the microchannel and the material of the microfluidic device [23,24]. The amplitude of acoustic waves can be increased by applying higher voltage or employing larger transducers, increasing the displacement of the piezoelectric element that induces the acoustic waves. However, the variability of these parameters is typically limited by spatial restrictions on the acoustofluidic device, heat generation from large electric inputs and damage of the device by a large displacement. The frequency of the acoustic waves is also important for efficient acoustic focusing, but is restricted by the dimensions of the focusing microchannel whose modification requires laborious procedures and sensitively affects the interaction with target particles. Furthermore, potential materials from which to fabricate microfluidic platforms are limited to hard materials such as glass, because of the need for acoustic waves to propagate efficiently through the substrate [23,24].

Despite the extensive considerations that have gone into analysis of important factors for the design of acoustic wave platforms, it has been overlooked how the dimensions of microfluidic devices, not the dimensions of microchannels, influence acoustic focusing. Our findings presented here indicate the importance of this investigation and will decisively benefit the future design of effective acoustofluidic devices.

A change in dimensions of a microfluidic device influences its acoustic focusing because the total mass of the device, being an oscillating object, is proportional to its volume. For example, decreasing the thickness (i.e. the distance between the top plane of the upper layer and the bottom plane of the lower layer) of a microfluidic device without changing the microchannel design itself readily increases the amplitude of acoustic waves and avoids the generation of additional heat otherwise caused by increasing the electric input into a piezoelectric transducer. Moreover, a change in device thickness does not change the conditions for particle measurements, except for their acoustic focusing.

In our set-up, we have used fixed planar dimensions, because changing the dimensions of planar surfaces of microfluidic devices is restricted by the microchannel design. In terms of device fabrication, the thickness of a microfluidic device is difficult to modify at the laboratory level, requiring special equipment and cumbersome fabrication processes.

We selected glass to use for our microfluidic devices because of its excellent optical transparency and its efficient propagation of acoustic waves, although there are other materials such as silicon and SU-8 that have been used to construct sophisticated acoustofluidic devices with good propagation of acoustic waves [25,26]. The transparency of glass allows us to obtain clear images under visible microscope. Furthermore, due to this transparency, glass is an ideal material for some measurement techniques such as Raman spectroscopy [27]. However, in the case of glass microfluidics, the modification of its thickness is challenging because this requires the flat surface to keep optical transparency. Thus, it is common that the thickness of a glass microfluidic device is equal to the thickness of the original glass component that was commercially available.

The thickness of a glass microfluidic device is an important parameter for acoustic focusing because it strongly correlates to the energy propagation of acoustic waves (see §2). The efficient propagation of acoustic energy is required, especially when it comes to acoustic focusing of micrometre-scale or smaller particles [23,26,28]. However, it has not yet been systematically discussed how the thickness of microfluidic devices influences the degree of acoustic focusing, because thin microfluidic devices are delicate to use and to produce. Thus, the thickness of microfluidic devices composed of hard materials is usually found to be 1–2 mm [29,30]. By contrast, we have worked on thinner glass

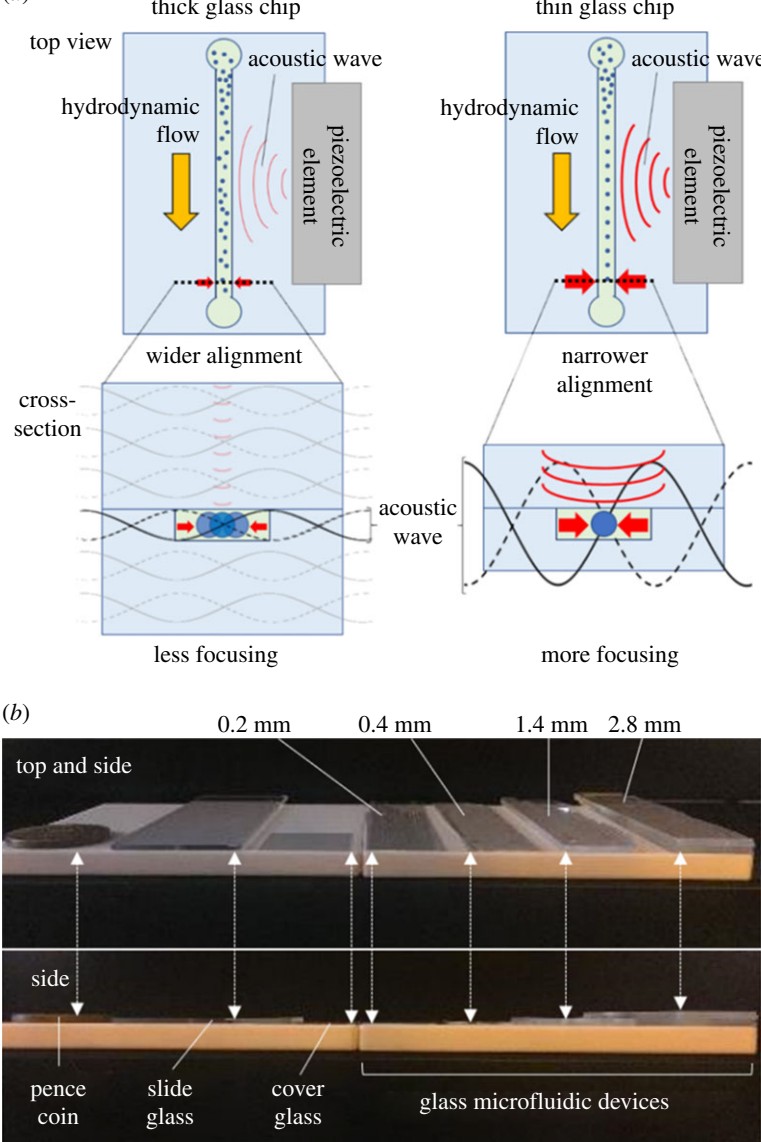

**Figure 1.** Acoustic waves and microfluidic devices of various thickness. (*a*) Acoustic focusing of particles in (left) thick and (right) thin microfluidic devices. The energy density of acoustic waves changes based on the total thickness of microfluidic devices. (*b*) Comparison of the thickness of glass microfluidic devices (2.8, 1.4, 0.4 and 0.2 mm thickness from right to left) with common objects (a pence coin, a slide glass and a cover glass (1.7, 1.0 and 0.15 mm thickness, respectively) from left to right). Each arrow indicates the same object.

microfluidic devices with a thickness as thin as 0.012 mm [31–35]. These devices were composed of only glass, fabricated via direct fusion-bonding of glass layers without an adhesive layer. This technique allowed us to investigate the correlation between the thickness of glass microfluidic devices and acoustic focusing.

When tiny particles are controlled by acoustic focusing, two types of forces affect particle alignment: the acoustic radiation force and the acoustic streaming force [29,36]. The acoustic streaming force is dominant for smaller particles (a few micrometres or less) and focuses particles in a curved trajectory [28,36,37]. The acoustic radiation force is, by contrast, dominant for relatively large particles and aligns particles in a straight line. However, previous studies report that submicrometre particles can also be aligned in a straight line by acoustic radiation when acoustic waves are effectively applied to these tiny particles [26]. Theoretically, it is expected that thin glass microfluidic devices can achieve straight alignment of tiny particles via acoustic radiation, because they can efficiently transduce acoustic waves to the microchannel. By contrast, this efficient propagation of the energy of acoustic waves is strongly limited in thick glass microfluidic devices (figure 1a).

We were able to overcome the limitations of previous fabrication techniques enabling us to experimentally verify this expectation by achieving straight alignment of small particles in thin glass microfluidic devices. Figure 1*b* shows the thin glass microfluidic devices that were used in this study and compares them with common objects.

For our investigation in influence of the thickness of microfluidic device on acoustic focusing, we firstly focused 2 µm beads through acoustic radiation. Then, a thin glass microfluidic device was employed to determine the degree of particle acoustic focusing by introducing micro/nanoparticles of various diameter ranging from 500 nm to 10 µm. Using thin glass microfluidic devices, we also demonstrate acoustic focusing of *Escherichia coli* as a model organism for small cells with diameters of up to a few micrometres.

# 2. Theory

Here, we show the mathematical description of how the thickness of a microfluidic device affects particle acoustic focusing by acoustic waves (figure 1*a*). Assuming that the material properties and intensity of acoustic waves are uniform in the device, the kinetic energy ($E$) of the microchip provided by applied acoustic waves is indicated in the following equation as the sum of the kinetic energy of atoms in simple harmonic oscillation in the material of the microfluidic device:

$$E = \frac{1}{2}m\bar{v}^2 = \frac{1}{2}m\bar{A}^2(2\pi f)^2 = 2\pi^2 m\bar{A}^2 f^2, \tag{2.1}$$

where $m$, $v$, $A$ and $f$ are the mass of the microchip, average velocity of the vibrating atoms, average amplitude of the oscillation and the frequency of the applied acoustic waves, respectively. If the energy introduced from outside (in this study via a piezoelectric transducer) is constant, and the loss of the energy transfer from the piezoelectric transducer to the microfluidic device is negligible, $mA^2f^2$ is considered to be constant. Since $f$ is fixed by the piezoelectric transducer, $A^2$ is considered to be inversely proportional to $m$. In the case of microfluidic devices with the same two-dimensional geometry, the mass of a microfluidic device is directly proportional to thickness of the device. Therefore, when the given energy is constant, the use of thin microfluidic devices results in an increased value for $A^2$, and thus, increased effect of acoustic focusing. In other words, realizing acoustic focusing in thin microfluidic devices requires smaller voltages when compared with thicker devices.

Next, we consider how a thin microfluidic device is advantageous over thick ones for acoustic focusing of small particles in straight alignment. When a particle is placed in an acoustic standing-wave field in a compressible-fluid medium, an acoustic radiation force ($F_r$) is induced by the existence of nonlinear terms for the vibration in the medium and it can push the particle towards the force-balanced point as given by the following equation [15,38]:

$$F_r = \frac{\pi p_0^2 V_p \beta_m}{2\lambda} \left( \frac{5\rho_p - 2\rho_m}{2\rho_p + \rho_m} - \frac{\beta_p}{\beta_m} \right) \sin(2kx), \tag{2.2}$$

where $p_0$ is the acoustic pressure, $V_p$ is the particle volume, $\lambda$ is the wavelength in the fluid medium, $k$ is the wavenumber, $x$ is the displacement along the wave propagation direction, $\rho_m$ is the density of the fluid medium, $\rho_p$ is the density of the particle, $\beta_m$ is the compressibility of the fluid medium and $\beta_p$ is the compressibility of the particle. From this equation, the acoustic radiation force is directly proportional to the particle volume, because the other parameter such as $\lambda$ is fixed by the channel geometry and material properties, respectively. To focus particles in smaller volumes, more acoustic pressure, which is proportional to the amplitude of standing waves, is required. Therefore, thinner glass microchips are more effective for acoustic focusing of particles.

# 3. Material and methods

## 3.1. Chemicals and materials

Deionized water was generated by Milli-Q Integral 15 (Merck Millipore, Billerica, MA, USA). The following chemicals of reagent grade were purchased from Wako Pure Chemical (Osaka, Japan): acetone, isopropanol, ethanol and HCl (35 wt%). Photoresist (OFPR-800) was purchased from Tokyo Ohka Kogyo (Tokyo, Japan). Hydrofluoric acid (HF, 49 wt%) was purchased from Morita Chemical Industries (Osaka, Japan). For glass etching, HF and HCl were diluted by deionized water to obtain

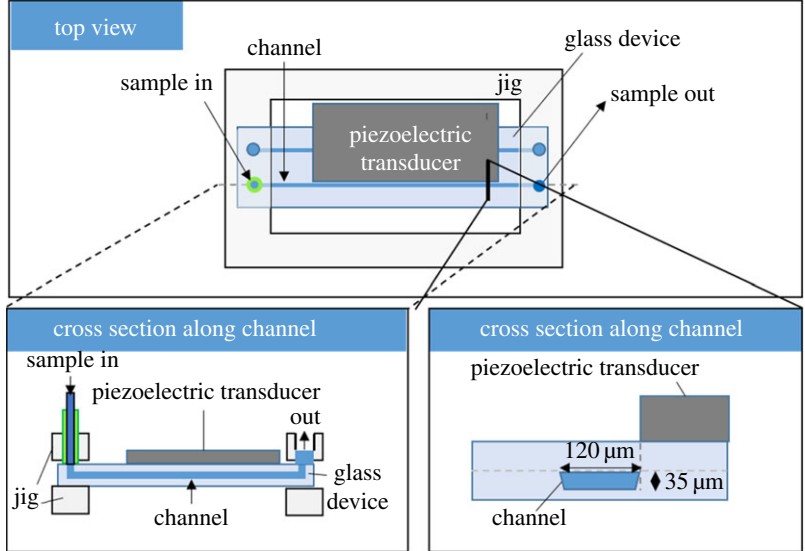

**Figure 2.** Illustration of a glass microfluidic device for acoustic focusing. A piezoelectric transducer was connected to a wave function generator for producing acoustic wave.

mixture of 20 wt% HF and 0.7 wt% HCl. Borosilicate glass plates of $70 \times 30$ mm with 0.1–0.7 mm thickness were purchased from Matsunami Glass (Osaka, Japan). The aluminium jig was manufactured by Institute of Microchemical Technology (Kanagawa, Japan).

Five sizes (0.5, 1, 2, 6 and 10 μm) of fluorescent polystyrene latex beads (Polyscience, Pennsylvania, PA, USA) were diluted in water for the experiments on acoustic focusing and enrichment of fluorescent particles. *Escherichia coli* (BL21) was purchased from BioDynamics Laboratory Inc. (Tokyo, Japan) and stored at $-80°C$. Before its utilization in acoustic focusing experiments, *E. coli* was defrosted to room temperature and cultured in 37°C incubator overnight.

## 3.2. Glass microfluidic device

The glass microfluidic device was fabricated by the previously described method [39] with slight modifications. In short, the glass plates were cleaned with $H_2SO_4/H_2O_2$ solution for 15 min to subsequently deposit Cr and Au layers on the glass surface by sputtering (EIS-220, Elionix, Tokyo, Japan). Photoresist (OFPR) was spin-coated on the metal layers and exposed to UV light for 14.5 s to develop the microchannel pattern. The pattern on the photomask had either a straight microchannel of 6 mm length and 50 μm width for acoustic focusing experiments or a channel of 5 mm length and 50 μm width that branched into a middle channel of 0.5 mm length and 1 μm width and two detours of 1.1 mm length and 140 μm width for the enrichment of fluorescent particles experiments. Through Cr, Au and glass etching by corresponding etchant, the microchannel of 35 μm depth was patterned onto the glass surface. The etched glass plate and a glass plate that had inlet and outlet ports made by a diamond-coated drill of 0.8 mm diameter (DCSSSD0080, Mitsubishi Materials, Tokyo, Japan) were treated with $H_2SO_4/H_2O_2$ solution to activate the glass surface. These activated glass plates were firstly bonded manually. For permanent bonding of the glass, a weight of 1.2 kg was placed on the top of them and the complex was exposed to a temperature of 620°C for 0.7 mm glass or 500°C for thinner glass. Then, the glass microfluidic device was cut to the dimensions of $70 \times 15$ ($H \times W$) mm, making it more sensitive to propagated acoustic waves.

## 3.3. Device set-up

To apply acoustic waves to a glass microfluidic device, a piezoelectric transducer ($40 \times 15$ mm, C-213, Fuji Ceramics, Shizuoka, Japan) was attached to the glass microfluidic device with cyanoacrylate glue (Aron Alpha 4000, Toagosei, Tokyo, Japan) along with a straight microchannel that had a length of either 50 mm (for the enrichment of particles experiments) or 60 mm (for the other experiments) (figure 2). The piezoelectric transducer was connected to a function generator (FY2300H, FeelTech, Henan, China) for actuation with a home-made power amplifier. The actuation frequency

for acoustic focusing of particles or cells on the nodal position was 3.98 MHz. The device was set on an aluminium jig under a microscope (IX-71, Olympus, Tokyo, Japan) for evaluation of particle acoustic focusing.

To investigate the difference in propagation of acoustic surface waves on glass and silicon, micro-vibrational amplitude of glass induced by acoustic waves was measured based on laser reflection. Both glass and silicon plates had thickness of 0.5 mm. As shown in electronic supplementary material, figure S1A, a He−Ne laser (632.8 nm, S145197, Lumentum, Milpitas, CA, USA) emitting a beam diameter of less than 1 mm was irradiated onto a specific point on the substrate. Four points at 15 mm intervals were investigated along the line of the attached piezoelectric transducer. The piezoelectric transducer was glued in place with cyanoacrylate glue and was connected to a function generator for actuation and a home-made power amplifier. The actuation frequency was 3.46 MHz by applying 20 V on the piezoelectric transducer to observe the largest amplitude for both glass and silicon. The reflected beam was collected by a segmented photodiode (SPOT 2D 1647-1, OSI Optoelectronics, Hawthorne, CA, USA). Fluctuation of the beam induced by micro-vibrations was changed to an electronic signal and amplified by instrumentation (BB-INA103, Texas Instruments, Dallas, TX, USA). The principle of this measurement system resembles that of an atomic force microscope, which is a highly accurate device to measure surface height in a non-contact manner. The difference of collected signals was directly proportional to the difference of the vibrational amplitudes.

## 3.4. Data collection

Fluorescent beads or *E. coli* were introduced at a set flow rate by a syringe pump (Fusion-400, Chemyx, Stafford, TX, USA). The piezoelectric transducer was vibrating at 3.98 MHz for observation of acoustic focusing under the microscope. Fluorescent beads were observed under a microscope (IX71, Olympus, Tokyo, Japan) with excitation/emission wavelengths of 480/520 nm and *E. coli* was screened by phase-contrast microscopy. The images were recorded through a CCD camera (TH4-100, Olympus) connected to a computer. The recorded images were analysed by the software ImageJ. For analysing the effects of glass thickness and bead size, collected images were merged into one picture to measure the dispersion of flowing beads with dark background. For analysis of how the beads could be concentrated by the device, beads on the obtained images were manually counted. For analysis of *E. coli* focusing, images were captured every 5 s to find the dispersion of flowing *E. coli*.

## 3.5. Computational analysis of fluid dynamics

Single-liquid-phase fluidic simulation was conducted by using the volume of fluid laminar flow model of branched microchannels to figure out the flow pattern, the velocity distribution and the possible trace of the beads. Simulation with the finite volume method was implemented by computational fluid dynamics software (Ansys 14.0, Ansys Inc., Canonsburg, PA, USA). Furthermore, for convenience of calculations, we used simple conditions of simulation in which a fixed wall and a steady inlet velocity were employed, and other influences were ignored. The obtained results are from the equilibrium state of the flow.

# 4. Results and discussion

## 4.1. Acoustic focusing experiments in glass microfluidic devices of various thicknesses

Compared with silicon, a frequently employed material for acoustofluidic devices, glass plate showed 10% smaller amplitude of propagated acoustic waves on average (electronic supplementary material, figure S1). Given that the density of silicon is 2.33 g cm$^{-3}$ and the density of borosilicate glass is 2.23 g cm$^{-3}$, the difference in the mass of the examined silicon and glass plates was 4.3%. According to equation (2.1) with these values, energy of the acoustic waves propagating on glass was 22% less than that on silicon. In other words, the 22% energy difference induced by the difference in device thickness corresponded to the material difference of glass and silicon. Therefore, this difference was small compared with the range of glass device thicknesses in this study.

Fluorescent beads of 2 µm diameter were introduced into a straight microchannel of 120 µm width and 35 µm depth at a flow rate of 0.2 µl min$^{-1}$ to investigate the correlation between acoustic focusing and the thickness of glass microfluidic devices. The channel width was empirically adjusted to 120 µm to avoid any disturbance, possibly due to inertial force, on acoustic focusing of particles [40].

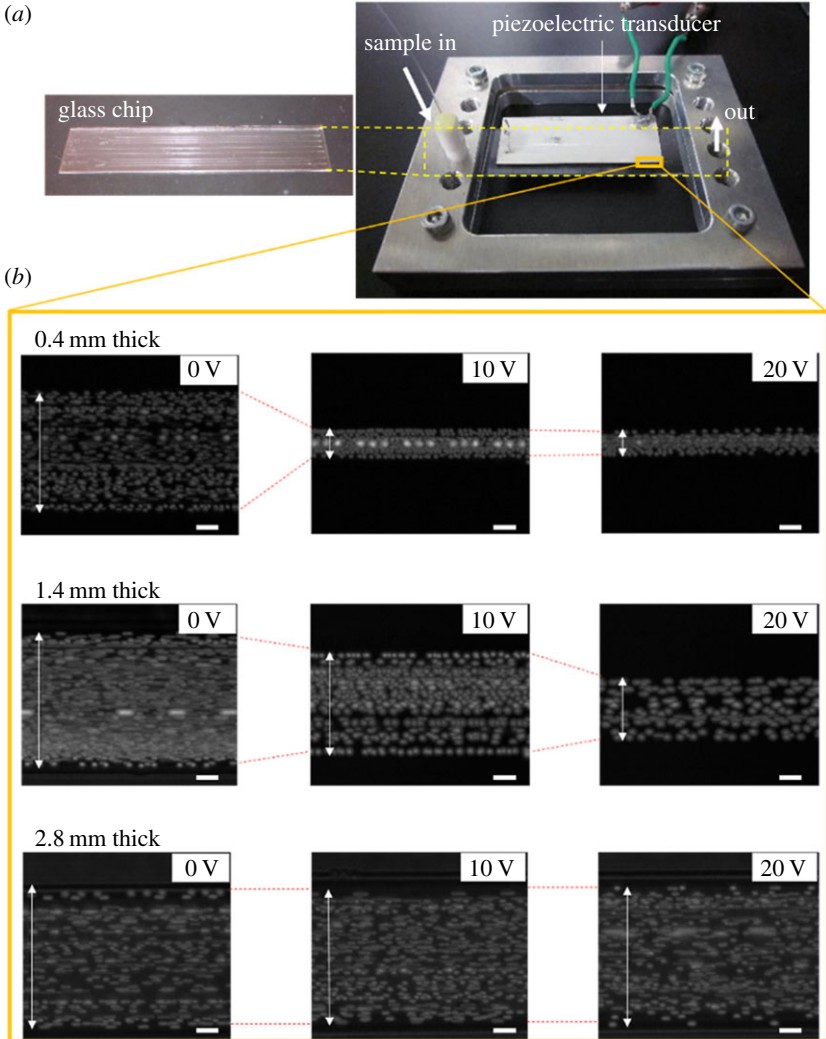

**Figure 3.** Device set-up and observation for acoustic focusing of 2.0 μm polystyrene beads. (*a*) A glass microfluidic device (left) and its set-up with a piezoelectric transducer on an aluminium jig (right). The observation spot of acoustic radiation is indicated on the device and (*b*) the degree of acoustic focusing was measured by dispersion of 2.0 μm beads (indicated by arrows) at 0, 10 or 20 V applied to the piezoelectric transducer on the glass microfluidic devices of 0.4, 1.4 or 2.8 mm thickness. Scale bar, 20 μm.

The thicknesses of the devices were 0.2, 0.4, 0.8, 1.4, 2.1 and 2.8 mm. Devices thinner than 0.2 mm were not used in these acoustic focusing experiments, because of their fragility under mechanical stress from acoustic waves (discussed later and in electronic supplementary material). Representing thicker microfluidic devices commonly used (1–2 mm), devices with a thickness of 1.4–2.8 mm were used to compare the achievable degree of acoustic focusing [29,30]. A piezoelectric transducer was attached to a glass microfluidic device along the length of the straight microchannel. The input voltage of the piezoelectric transducer ranged from 0 to 20 V in 2 V increments. The acoustic focusing of fluorescent beads was evaluated near the outlet edge of piezoelectric transducer (figure 3*a*) and observed images were merged to measure the dispersion of flowing particles.

It was shown that fluorescent beads were clearly focused and aligned straight in 0.4 mm glass microfluidic devices by applying 10 V or more to the piezoelectric transducer, indicating that the acoustic radiation force was dominant over acoustic streaming force (figure 3*b*). Meanwhile, applying the same scheme to thicker devices resulted in wider dispersion of flowing beads, or no acoustic focusing in devices of 1.4 and 2.8 mm thickness, respectively (figure 3*b*). These results are summarized in figure 4, and the normalized dispersion was calculated based on the dispersion of beads at 0 V (i.e. under no acoustic focusing). The results indicate that acoustic focusing is clearly influenced by the thickness of glass microfluidic devices. In devices of 0.2–0.8 mm thickness, flowing beads were focused by applying 10 V or less. By applying 16 V or more, dispersion of flowing beads was in similar degree, approximately 20–30% of maximum dispersion that was found at 0 V. By

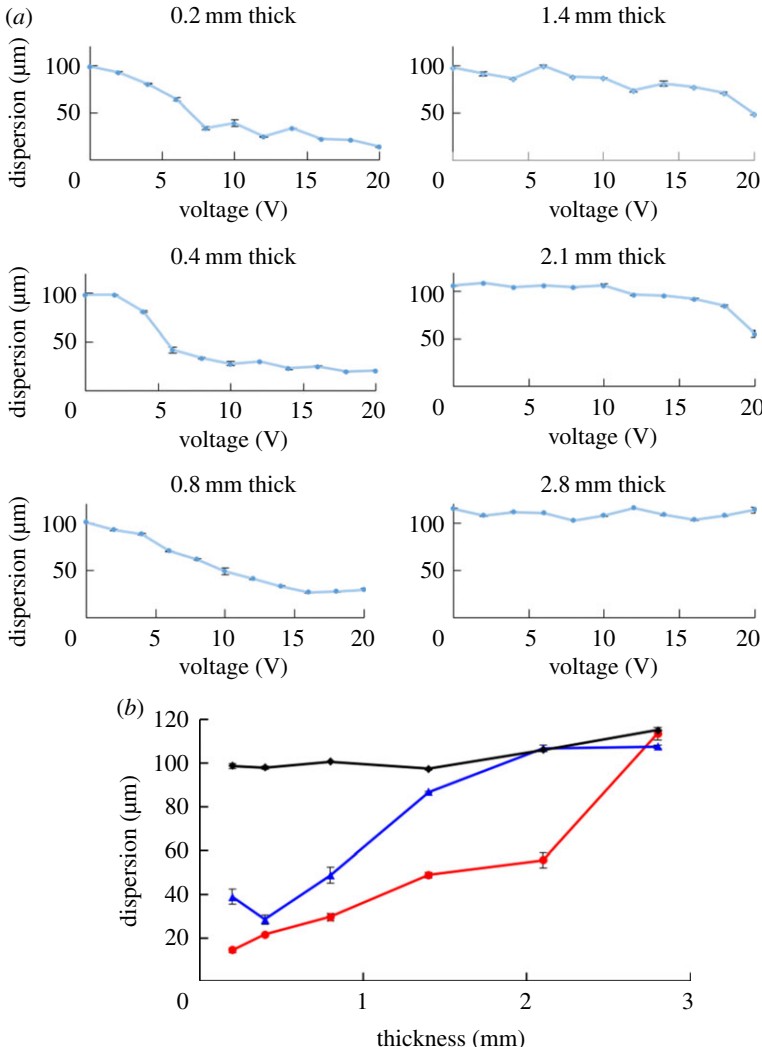

**Figure 4.** Acoustic focusing influenced by thickness of microfluidic devices. (*a*) Average dispersion of 2.0 μm beads in glass microfluidic devices of various thickness under influence of acoustic focusing by applying 0–20 V to a piezoelectric transducer (*n* = 3 for each data point with standard error) and (*b*) summary of average dispersion of 2.0 μm beads at various thicknesses. Dispersion of the beads was measured by applying 0 V (black diamond), 10 V (blue triangle) or 20 V (red square) to a piezoelectric transducer. Error bars show standard error (*n* = 3).

contrast, devices of 1.4 and 2.1 mm thickness showed apparent acoustic focusing of flowing beads only when the applied voltage was close to 20 V. A device of 2.8 mm thickness did not show clear acoustic focusing.

In further acoustic focusing experiments, we investigated ultrathin glass microfluidic devices of 0.012 mm thickness to check whether it was feasible to find the degree of acoustic focusing in a device with extreme thickness. This glass device was composed of three glass layers of 0.004 mm thickness (electronic supplementary material, figure S2a) with microchannel dimensions of 4 μm height and 120 μm width. We fabricated the channel layer using a femtosecond laser and sandwiched it between an orifice layer and a bottom layer using the previously described method [31] (electronic supplementary material, figures S2b and c). Then, the three layers (electronic supplementary material, figure S2d) were fusion-bonded (electronic supplementary material, figure S2e and f) and a piezoelectric transducer was attached.

We used beads with a 280 nm diameter (FP-0262, Spherotech Inc., Lake Forest, IL, USA) in these acoustic focusing experiments. Applying up to 20 V to the attached piezoelectric transducer did not lead to acoustic focusing of the beads (electronic supplementary material, figure S3b and c). For such a thin chip, the mechanical stress induced by acoustic waves consistently lead to cracking of the device, making it unfeasible for acoustic focusing. In most cases, the generated acoustic wave cracked the ultrathin device (electronic supplementary material, figure S3d and e). Based on the results

obtained with microfluidic devices of 0.012–2.8 mm thickness, devices with a thickness of 0.4 mm exhibit the best balance between acoustic focusing capability and durability.

Overall, by establishing a practical method for fabrication of thin microfluidic devices, we experimentally investigated how thin glass microfluidic devices are suitable for acoustic focusing of flowing beads, as expected from theoretical calculations. In general, the results follow two trends: (i) for the same applied voltage, a higher degree of acoustic focusing of beads was observed in thinner microfluidic devices, and (ii) the degree of focusing in a device is positively correlated to the strength of the incoming acoustic waves (i.e. higher voltage applied to piezoelectric transducer). Based on these experimental results, we conclude that thinner glass microchips exhibit superior properties for narrow focusing of particles being able to propagate larger energies of acoustic waves to flowing beads.

## 4.2. Correlation between acoustic focusing and diameter of polystyrene beads in a glass microfluidic device of 0.4 mm thickness

Fluorescent beads of 0.5, 1, 2, 6 or 10 μm diameter were introduced into a straight microchannel of 120 μm width and 35 μm depth in a microfluidic device of 0.4 or 2.1 mm thickness at a flow rate of 0.2 μl min$^{-1}$. Again, the dispersion of flowing particles was measured by recording with a CCD camera. As shown in figure 5a, applying 20 V to the piezoelectric transducer, 0.5 μm beads were focused only in thinner (0.4 mm) devices, while 2 μm beads were focused in both 0.4 and 2.1 mm devices, although a narrower dispersion was found in the 0.4 mm device. Further acoustic focusing results have been collected and summarized in figure 5b.

In general, fluorescent beads of larger diameter were focused narrowly in glass microfluidic devices of 0.4 and 2.1 mm thickness, although the beads of each diameter appeared to be less dispersed in a 0.4 mm device when compared with a 2.1 mm device. The strongest acoustic focusing was observed for 10 μm beads in the 0.4 mm device, while the amplitude of acoustic waves or diameter of beads needed to be larger to attain single-line focusing. The results indicate that acoustic focusing of beads with a diameter of 1 μm or smaller was critically influenced by the thickness of the glass microfluidic device (figure 5b), while fluorescent beads with a diameter of 2 μm or larger could be focused to 30% or less of the maximum dispersion at 0 V in both 0.4 and 2.1 mm glass microfluidic devices. Comparing this result with previous reports on acoustic focusing for 0.6 μm particles with a normalized dispersion of 0.48 [26], the present study achieves a narrower normalized dispersion (0.28) for 0.5 μm particles. While the present study applies higher voltage to a transducer (20 versus 11 V) with different frequencies of acoustic waves (3.98 versus 3.24 MHz), this difference could be caused by differences in the width (120 versus 230 μm) and device material (glass versus glass and silicon) to induce effective acoustic focusing. Furthermore, considering different diameters of particles (0.5 versus 0.6 μm), acoustic radiation forces in this study are almost comparable from equations (2.1) and (2.2) if the difference in thickness of the microfluidic devices is neglected. Therefore, the degree of acoustic focusing might be higher in this study. This would be due to the use of a thin glass microfluidic device, although thickness of the microfluidic device in the previous study [26] was unknown. At least, our results on particle acoustic focusing could be reasonably close to the results of the previous study.

Overall, thin glass microfluidic devices were found to enhance focusing of tiny particles, including submicrometre particles that could not be focused effectively in thicker glass microfluidic devices. In conclusion, changing the thickness of microfluidic devices has been shown to be an option to enhance acoustic focusing besides modifying the parameters of piezoelectric transducers or dimensions of the microchannel. Future designs of microfluidic devices are expected to considerably benefit from consideration of the devices' thickness.

## 4.3. Designing a thin glass microfluidic device based on simulation for enrichment of polystyrene beads with acoustic focusing

As an application of acoustic focusing, a thin glass microfluidic device was employed for enrichment of small particles. Since acoustic focusing can confine particles in space, it works as a powerful technique to collect target particles while undesired objects are excluded. For example, high-speed cytometry requires enrichment of cells because large volumes of sheath liquid are used to flow cells [41,42]. This usually results in a lowered concentration of cells and hampers identification of target cells.

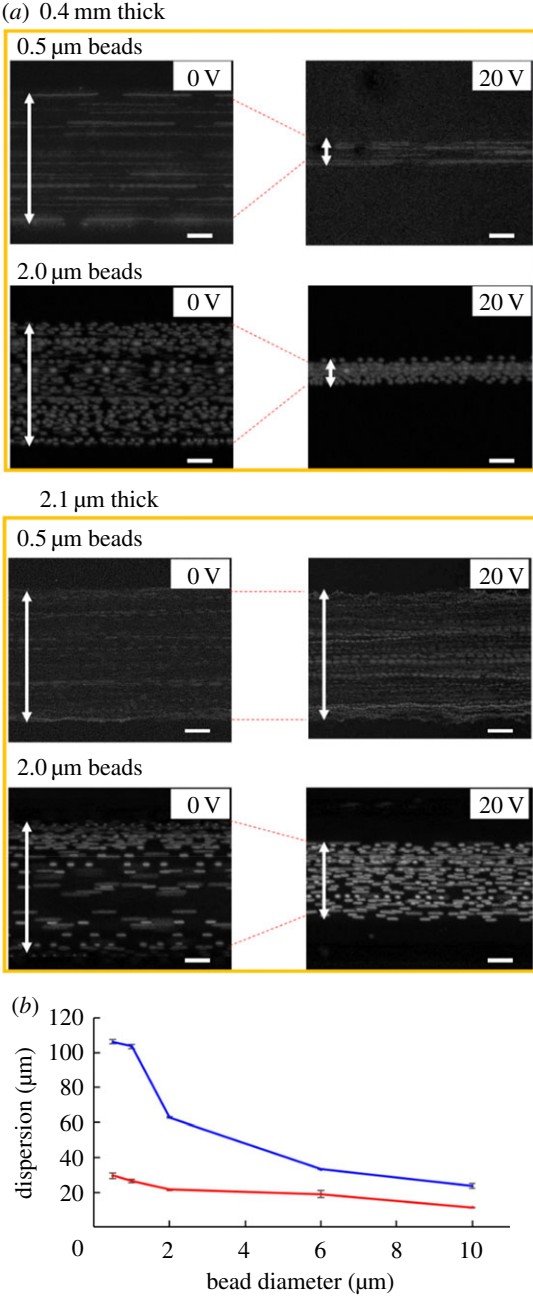

**Figure 5.** Acoustic focusing of polystyrene beads of various sizes in a thin (0.4 mm) and a thick (2.1 mm) microfluidic device. (a) Dispersion of 0.5 or 2.0 μm beads (indicated by arrows) in a microfluidic device of 0.4 or 2.1 mm thickness by applying 0 or 20 V to a piezoelectric transducer. Scale bar, 20 μm. (b) Summary of dispersion of 0.5 – 10 μm beads in a microfluidic device of 0.4 or 2.1 mm thickness by applying 20 V to a piezoelectric element. Circle (red): 0.4 mm thickness; square (blue): 2.1 mm thickness. Error bars show the standard error ($n = 3$).

However, by applying acoustic focusing, unnecessary liquid can be discarded and the concentration of cells recovered [43].

To investigate the effect of acoustic focusing on the enrichment of target particles, a glass microfluidic device of 0.4 mm thickness was designed based on computational fluid dynamics analysis. In the numerical flow simulation, the simulation domain included a channel of 120 μm width that branched into a middle channel of 70 μm width and two detours of 240 μm width (electronic supplementary material, figure S4a). The 35 μm height of the microchannel was uniform. The outlet boundaries of the middle channel and the two detours were set as pressure outlet. The inlet boundary of the 120 μm channel was set as the flow-velocity inlet (electronic supplementary material, figure S4b). The modelled domain was meshed into small cubes with finite volume. The physical properties of

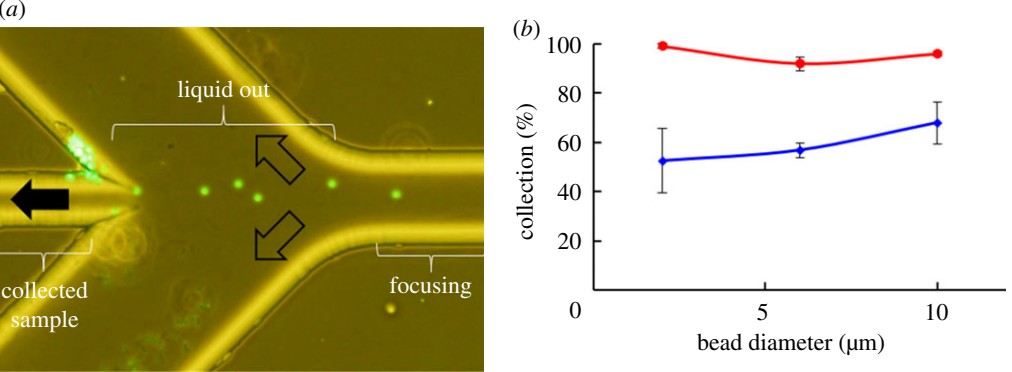

**Figure 6.** Enrichment of polystyrene beads in a branched channel. Enrichment was conducted in a glass microfluidic device of 0.4 or 1.8 mm thickness by applying 20 V to the piezoelectric transducer. (*a*) The part of the glass chip where beads were collected. The narrow channel in the middle left is for particle collection (flow direction is indicated by a solid arrow) and broad channels at the upper and lower left are for draining excess liquid (flow directions are indicated by outlined arrows). (*b*) Summary of successful collection of 2, 6 and 10 μm beads in 0.4 (red circle) or 1.8 mm (blue square) thick glass devices. Each error bar represents the standard error (*n* = 3).

the upper, side and bottom solid surfaces were set to those of glass with non-slip boundary conditions. The inlet boundary condition was given as the velocity inlet. The temperature was set to 25°C. The physical properties of the fluid were set to those of pure water; the viscosity $\mu$ was set to 0.653 mPa s and the density $\rho$ to $1.02 \times 10^3$ kg m$^{-3}$. The flow velocity was adjusted to 1.92 and 3.9 mm s$^{-1}$ (0.5 and 1 μl min$^{-1}$). Through computational simulations, the flow streamlines were obtained and approximately one-tenth of introduced liquid is collected through the 70 μm middle channel (figure 6*b*). The simulated flow streams indicate that targets (i.e. fluorescent particles) focused into the expected focus area (in the centre of the 120 μm channel, width of less than 25 μm) can be collected through the 70 μm middle channel and concentrated tenfold. Based on these results, a glass microfluidic device with a branched microchannel was fabricated by the wet etching process described in §3.2.

For enrichment of fluorescent polystyrene beads, a piezoelectric transducer was attached to the glass microfluidic channel alongside the straight microchannel. Fluorescent beads were introduced into the microchannel at a flow rate of 0.5 μl min$^{-1}$ for beads of 2 or 6 μm diameter or 1 μl min$^{-1}$ for beads of 10 μm diameter. The beads were counted where the microchannel branched to the middle channel and the detours to find the effect of device thickness on particle enrichment before being affected by different dimensions of the middle channel and the detour. The introduced beads were focused by acoustic radiation through a straight microchannel and collected through the middle channel in the 0.4 mm device (electronic supplementary material, Movie S1). Even though it was not optimal focusing to attain collection of all beads in the 0.4 mm device, acoustic focusing was much stronger in the 0.4 mm device than in the 1.8 mm device (electronic supplementary material, Movie S2). A portion of the liquid that suspended fluorescent beads flowed into the detours as waste. Thus, particles were concentrated by flowing through this microfluidic device when they flowed into the middle channel for collection (figure 6*a*).

Applying acoustic focusing with 20 V input voltage to the glass device, more than 90% of the beads were collected successfully in the middle channel for all bead sizes (figure 6*b*). Meanwhile, the rate of successful collection was lowered when no acoustic wave was applied to the glass device. These results indicate that acoustic focusing helps concentrating particles by positioning beads in the centre of the straight channel. This type of setting can be applied to collecting particles, for example, rare cells in large amounts from culture media or body fluid.

## 4.4. Thin glass microfluidic device for *Escherichia coli* acoustic focusing experiments

Acoustic focusing is also applicable for biological particles such as microorganisms. To characterize properties of these particles such as size and shape, acoustic focusing helps to obtain crisp images by confining these particles in narrow space. In continuous flow, straight alignment of particles by acoustic radiation is desired.

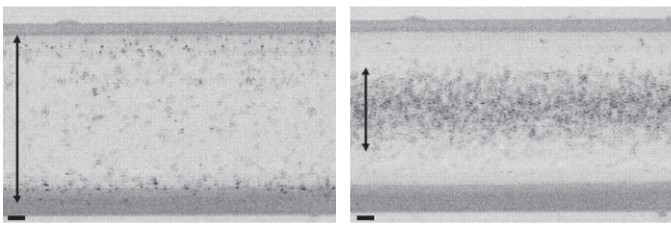

**Figure 7.** Acoustic focusing of *E. coli* in a glass chip of 0.4 mm thickness by applying (left) 0 or (right) 20 V to a piezoelectric transducer. Each image was obtained by merging 10 pictures (obtained in every 5 s) with background subtraction. The average dispersion of *E. coli* with standard error ($n = 10$) was found (left) $103.1 \pm 2.1$ and (right) $52.8 \pm 4.2$ $\mu$m, respectively, and indicated by arrows. Scale bar, 20 $\mu$m.

To investigate the applicability of acoustic focusing with thin glass microchips to tiny biological particles, we selected *E. coli* as a model organism with a diameter around 1 $\mu$m [26]. In the acoustic focusing experiments, *E. coli* in culture media were directly introduced to a straight microchannel of 120 $\mu$m width and 35 $\mu$m depth in the 0.4 mm glass microfluidic device at flow rate of 0.2 $\mu$l min$^{-1}$. Acoustic focusing of *E. coli* was evaluated near the outlet edge of the piezoelectric transducer. Applying acoustic waves (20 V input) to the glass device, an average dispersion of *E. coli* with standard error was found to be $52.8 \pm 4.2$ $\mu$m ($n = 10$) when acoustic focusing was applied, while the average dispersion of *E. coli* with standard error increased to $103.1 \pm 2.1$ $\mu$m ($n = 10$), which is close to the width of the microchannel, when no acoustic wave was applied. Thus, acoustic waves focused *E. coli* in 51.2% of the unfocused dispersion in the 0.4 mm glass microfluidic device. In figure 7, merged pictures of acoustically focused and unfocused *E. coli* are provided for comparison. Also, we found that *E. coli* was active before and after flowing through the microchannel with acoustic focusing (electronic supplementary material, figure S5 and movies S3 and S4). These results match previous reports on the viability of cells before and after exposure to acoustic waves [21,22].

These results suggest that acoustic focusing will also be applicable to microorganisms of a few micrometres in thin glass microfluidic devices, even though our device was not optimized for this focusing and this field still requires further scientific development. As current technologies employ acoustic focusing for enhancing screening and analysis of microorganisms in flow systems such as flow cytometry, thin glass microfluidic devices offer new possibilities to further enhance the efficiency of acoustic focusing systems.

# 5. Conclusion

In this study, we used glass acoustofluidic devices to investigate how acoustic focusing of flowing particles was influenced by the thickness of the employed glass microfluidic devices. Thin glass microfluidic devices significantly enhanced focusing of small particles by acoustic radiation force, due to their highly efficient energy propagation of acoustic waves to target particles. Although acoustic waves propagated 10% smaller amplitude on glass than silicon, glass was suitable for our investigation because glass microfluidic devices could be produced without intermediate layer, adhesive materials or specialized apparatus. In this set-up, we demonstrated that acoustic focusing is applicable to concentrating particles by removal of excess liquid, and to acoustic focusing of *Escherichia coli* of one to a few micrometre in size.

Using thin glass microfluidic devices, it is possible to adopt more diverse layouts of microchannels that may not be advantageous in terms of the energy propagation of acoustic waves but are desirable in special cases, e.g. when using shallow microchannels for observation of particles in fast flow [40]. Possible applications that thin glass microfluidic devices can contribute to are the isolation and analysis of rare cells and organelles with imaging [40,44,45], as well as separating mixed particles with diverse sizes [46,47].

Although a future investigation is desired to study correlations between acoustic focusing and thickness of microfluidic devices composed of materials other than glass, the acoustic focusing capability is strongly dependent on the thickness of the microfluidic devices composed of glass as theoretically expected. Engineering the device thickness as a method to enhance acoustic focusing does not require a change of equipment or microchannel dimensions, effectively eliminating the two major practical difficulties that hamper high acoustic focusing efficiencies in current set-ups. Finally,

we expect thin glass microfluidic devices to become an essential component in future microfluidic systems that require optimized acoustic focusing.

Data accessibility. Supporting data for this article have been uploaded as part of the electronic supplementary material. Electronic supplementary material contains fabrication process of the 0.012 mm glass microfluidic device and the experiments on acoustic focusing, enrichment of fluorescent particles and acoustic focusing on *E. coli*.

Authors' contributions. All authors contributed to designing the research plan and instrumental set-up and optimization. N.O., Y.Y. and T.S. performed the experiments of acoustic focusing and simulations. The manuscript was written by N.O. and Y.Y. and the others critically revised the manuscript. All the authors discussed and commented on the manuscript.

Competing interests. The authors have no competing interests.

Funding. This research was funded by the ImPACT Program of the Council for Science, Technology and Innovation (Cabinet Office, Government of Japan).

Acknowledgements. The authors thank Dr Katsuhiko Matsumoto for his help on *E. coli* treatment and Dr Nobuyuki Tanaka for useful discussions on data analysis.

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
