## [Reviewer comments · Royal Society Open Science]

Review History

RSOS-180731.R0 (Original submission)

Review form: Reviewer 1

Is the manuscript scientifically sound in its present form?

No

Are the interpretations and conclusions justified by the results?

No

Is the language acceptable?

No

Is it clear how to access all supporting data?

Yes

Do you have any ethical concerns with this paper?

No

Have you any concerns about statistical analyses in this paper?

No

Recommendation?

Reject

Comments to the Author(s)

Comments:

Authors tried to discuss an essential aspect of acoustophoretic devices which is of clear importance to the acoustofluidic community. However, I am not convinced with the conclusion that the authors made based on the reported data and discussion. It is obvious that the thickness of acoustic devices has an impact on the efficiency of focusing and a study like this should do a thorough investigation without limiting to one type of material as stated below. Overall, the manuscript contains significant weaknesses, therefore, I cannot recommend publishing the manuscript in the current format.

1) Authors used only glass as a substrate to fabricate microfluidic device but, silicon is the most commonly used substrate in bulk acoustic wave-based devices. Glass substrates are rarely used thus I am not convinced about the study just limiting to the glass. Silicon has better acoustic properties than glass. It would make the manuscript convincing if authors compare the Si-devices with different thicknesses and/or different thicknesses of top glass layer which is bonded to Si on such devices. Silicon devices can easily be integrated with optical detection systems by having a glass lid; thus I cannot accept the reason to use glass only devices in this study.

2) Page 4 line 48, the authors mentioned the flow rate used is 0.2 $\mu\text{L}/\text{min}$, which is a very low flow rate in common acoustophoresis experiments. Effect of thickness is not described in higher flow rates such as 100 $\mu\text{L}/\text{min}$. The supporting movies indicate that even at this very low flow rate the extent of focusing is poor. Movie 2 clearly shows that particles are not focused and they are directed into three outlets. Previous work from other groups has utilized high flow rates with thick devices yet with very efficient focusing.

3) Authors mentioned that they have used 120 μm wide channels. Assuming the device was operated at half wavelength mode, the resonance frequency used (3.98 MHz) does not match with the width (assuming the speed of sound in the medium ~ 1500 m/s).

4) Page 5, Line 51-52: Authors reasoned that they had to use high voltage due to high frequency (3.98 MHz). However, particles should focus at a lower voltage when the resonance frequency is high.

5) Design and fabrication of the device are not discussed well in the manuscript.

6) Authors claim that thinner glass substrates ($<0.2\text{mm}$) are not useful due to the fragility under acoustic mechanical stress. I am not sure why they still investigated devices with 12 μm thickness. There is no use of such devices for real applications. (e.g. Pressure drop issues and particle clogging will be significant)

7) The extent of the focusing is calculated based on the distribution of particles inside the microchannels, but no quantitative measurement of the samples collected from each outlet is provided.

8) On page 5 line 17 authors discuss using 280 nm particles for the acoustic focusing experiment. No matter the thickness of the device, it is well accepted fact that manipulation of submicron particles with direct acoustophoresis is really hard. This adds up nothing to the manuscript. And acoustic streaming effects can be significant for nanoparticles at very slow flow rates.

9) In usual acoustophoresis setup, 20 Vpp is sufficient to focus the particles $>2 \mu\text{m}$ in size in a very narrow space within the microchannel. Authors do not mention why it is not the case with glass devices.

10) Figures 3b and 5b must have scale bars

11) The manuscript has some narration inconsistencies and typos.

Review form: Reviewer 2

Is the manuscript scientifically sound in its present form?

Yes

Are the interpretations and conclusions justified by the results?

Yes

Is the language acceptable?

Yes

Is it clear how to access all supporting data?

Yes

Do you have any ethical concerns with this paper?

No

Have you any concerns about statistical analyses in this paper?

No

Recommendation?

Major revision is needed (please make suggestions in comments)

Comments to the Author(s)

The authors considered the effects of glass thickness on acoustic particle focusing. Although the obtained results are limited to glass microfluidic devices, I still think the work is interesting and may provide some guidelines for designing glass-based acoustic microfluidics. In their future works, the authors are strongly encouraged to explore this thickness-effect in devices of other popular materials (e.g., PDMS, SU-8). Overall, I would like to recommend its publication in Royal Society Open Science if the authors are willing to address the following comments.

(1) I don't know the meaning of "the thickness of a glass microfluidic device" (i.e., page 2, line 29). The authors are suggested to give a clear definition. Does it mean the distance between the transducer and the microchannel or the thickness of the microchannel in the bulk glass?

(2) The simulation of the flow in branched microchannel is very simple and meaningless. It would be better if the authors move it to the supporting information.

(3) The focusing performance under the optimal condition is not very good. Is it possible to

achieve the single-line focusing using this acoustic focusing device?

(4) In the introduction, the authors are suggested to briefly discuss the importance of particle focusing (e.g., for particle/cell separation (*Analytical chemistry*, 2018, 90(6): 4212-4220.) or microflow cytometer (*Sensors and Actuators B: Chemical*, 2018, 266:26-45; *Analytical chemistry*, 2017, 89(5): 3154-3161.)). In addition, the other active or passive focusing schemes (e.g., inertial microfluidic, electric, magnetic and optical methods) are suggested to be discussed. This would make the readers better understand the meaning of studying particle focusing and why the acoustic particle focusing is studied in this work.

Decision letter (RSOS-180731.R0)

05-Sep-2018

Dear Dr Tanaka:

Manuscript ID RSOS-180731 entitled "Enhancement in acoustic focusing of micro/nanoparticles by thinning a microfluidic device" which you submitted to Royal Society Open Science, has been reviewed. The comments from reviewers are included at the bottom of this letter.

In view of the criticisms of the reviewers, the manuscript has been rejected in its current form. However, a new manuscript may be submitted which takes into consideration these comments.

Please note that resubmitting your manuscript does not guarantee eventual acceptance, and that your resubmission will be subject to peer review before a decision is made.

Your resubmitted manuscript should be submitted by 05-Mar-2019. If you are unable to submit by this date please contact the Editorial Office.

Please note that Royal Society Open Science will introduce article processing charges for all new submissions received from 1 January 2018. Charges will also apply to papers transferred to Royal Society Open Science from other Royal Society Publishing journals, as well as papers submitted as part of our collaboration with the Royal Society of Chemistry (<http://rsos.royalsocietypublishing.org/chemistry>). If your manuscript is submitted and accepted for publication after 1 Jan 2018, you will be asked to pay the article processing charge, unless you request a waiver and this is approved by Royal Society Publishing. You can find out more about the charges at <http://rsos.royalsocietypublishing.org/page/charges>. Should you have any queries, please contact openscience@royalsociety.org.

Kind regards,
Royal Society Open Science Editorial Office

on behalf of Dr Chong Li (Associate Editor) and Prof. R. Kerry Rowe (Subject Editor)
openscience@royalsociety.org

Associate Editor Comments to Author (Dr Chong Li):

Thank you for submitting your work to Royal Society Open Science for consideration of publishing. Unfortunately we regret to reject it as its current form. Based on the feedback from our fellow reviewers, there are major defects in the paper although significant amount of efforts have been spent on investigating the effects of glass thickness on acoustic particle focusing. Please refer to the reviewer's comments for details when considering to resubmit the paper.

Reviewers' Comments to Author:

Reviewer: 1

Comments to the Author(s)

Comments:

Authors tried to discuss an essential aspect of acoustophoretic devices which is of clear importance to the acoustofluidic community. However, I am not convinced with the conclusion that the authors made based on the reported data and discussion. It is obvious that the thickness of acoustic devices has an impact on the efficiency of focusing and a study like this should do a thorough investigation without limiting to one type of material as stated below. Overall, the manuscript contains significant weaknesses, therefore, I cannot recommend publishing the manuscript in the current format.

1) Authors used only glass as a substrate to fabricate microfluidic device but, silicon is the most commonly used substrate in bulk acoustic wave-based devices. Glass substrates are rarely used thus I am not convinced about the study just limiting to the glass. Silicon has better acoustic properties than glass. It would make the manuscript convincing if authors compare the Si-devices with different thicknesses and/or different thicknesses of top glass layer which is bonded to Si on such devices. Silicon devices can easily be integrated with optical detection systems by having a glass lid; thus I cannot accept the reason to use glass only devices in this study.

2) Page 4 line 48, the authors mentioned the flow rate used is 0.2 $\mu\text{L}/\text{min}$, which is a very low flow rate in common acoustophoresis experiments. Effect of thickness is not described in higher flow rates such as 100 $\mu\text{L}/\text{min}$. The supporting movies indicate that even at this very low flow rate the extent of focusing is poor. Movie 2 clearly shows that particles are not focused and they are directed into three outlets. Previous work from other groups has utilized high flow rates with thick devices yet with very efficient focusing.

3) Authors mentioned that they have used 120 μm wide channels. Assuming the device was operated at half wavelength mode, the resonance frequency used (3.98 MHz) does not match with the width (assuming the speed of sound in the medium ~ 1500 m/s).

4) Page 5, Line 51-52: Authors reasoned that they had to use high voltage due to high frequency (3.98 MHz). However, particles should focus at a lower voltage when the resonance frequency is high.

5) Design and fabrication of the device are not discussed well in the manuscript.

- 6) Authors claim that thinner glass substrates (<0.2mm) are not useful due to the fragility under acoustic mechanical stress. I am not sure why they still investigated devices with 12 μm thickness. There is no use of such devices for real applications. (e.g. Pressure drop issues and particle clogging will be significant)
- 7) The extent of the focusing is calculated based on the distribution of particles inside the microchannels, but no quantitative measurement of the samples collected from each outlet is provided.
- 8) On page 5 line 17 authors discuss using 280 nm particles for the acoustic focusing experiment. No matter the thickness of the device, it is well accepted fact that manipulation of submicron particles with direct acoustophoresis is really hard. This adds up nothing to the manuscript. And acoustic streaming effects can be significant for nanoparticles at very slow flow rates.
- 9) In usual acoustophoresis setup, 20 Vpp is sufficient to focus the particles >2 μm in size in a very narrow space within the microchannel. Authors do not mention why it is not the case with glass devices.
- 10) Figures 3b and 5b must have scale bars
- 11) The manuscript has some narration inconsistencies and typos.

Reviewer: 2

Comments to the Author(s)

The authors considered the effects of glass thickness on acoustic particle focusing. Although the obtained results are limited to glass microfluidic devices, I still think the work is interesting and may provide some guidelines for designing glass-based acoustic microfluidics. In their future works, the authors are strongly encouraged to explore this thickness-effect in devices of other popular materials (e.g., PDMS, SU-8). Overall, I would like to recommend its publication in Royal Society Open Science if the authors are willing to address the following comments.

- (1) I don't know the meaning of "the thickness of a glass microfluidic device" (i.e., page 2, line 29). The authors are suggested to give a clear definition. Does it mean the distance between the transducer and the microchannel or the thickness of the microchannel in the bulk glass?
- (2) The simulation of the flow in branched microchannel is very simple and meaningless. It would be better if the authors move it to the supporting information.
- (3) The focusing performance under the optimal condition is not very good. Is it possible to achieve the single-line focusing using this acoustic focusing device?
- (4) In the introduction, the authors are suggested to briefly discuss the importance of particle focusing (e.g., for particle/cell separation (*Analytical chemistry*, 2018, 90(6): 4212-4220.) or microflow cytometer (*Sensors and Actuators B: Chemical*, 2018, 266:26-45; *Analytical chemistry*, 2017, 89(5): 3154-3161.)). In addition, the other active or passive focusing schemes (e.g., inertial microfluidic, electric, magnetic and optical methods) are suggested to be discussed. This would make the readers better understand the meaning of studying particle focusing and why the acoustic particle focusing is studied in this work.

Author's Response to Decision Letter for (RSOS-180731.R0)

See Appendix A.

RSOS-181776.R0

Review form: Reviewer 1

Is the manuscript scientifically sound in its present form?

No

Are the interpretations and conclusions justified by the results?

No

Is the language acceptable?

Yes

Is it clear how to access all supporting data?

Yes

Do you have any ethical concerns with this paper?

No

Have you any concerns about statistical analyses in this paper?

No

Recommendation?

Major revision is needed (please make suggestions in comments)

Comments to the Author(s)

In this resubmission, authors have tried to address my main comments. Authors have mentioned the reason for using the glass-glass device instead of the more commonly used glass-silicon device; however, I am still not convinced by the performance of this device. Authors have failed to demonstrate good focusing in any of the devices for 10 μm particles even at very slow flow rates (0.2 $\mu\text{L}/\text{min}$). There is so many previous works showing excellent focusing of particles and cells smaller than 10 μm at much higher flow rates than 0.2 $\mu\text{L}/\text{min}$. Since authors have not investigated the most common device type (glass-silicon), to come to a conclusion and use the finding for further studies, authors should at least first establish a working device. Authors have mentioned the reason for using slow flow rate was due to the weak focusing of sub-micron particles. It is not necessary to use sub micron-particles to establish the phenomenon. Acoustic streaming can interfere with particle focusing when the particles are smaller than 1 μm , and the flow rate is very slow. The observation in Movie 1 cannot be considered as good focusing. The end of Movie 1 shows some non-focused particles. The movies are so short that it is difficult to be convinced that the device is capable of good focusing. I would like to see a more extended video clip with unfocused stream getting focused in real time. Also, the video should be collected away from the trifurcate end so that the focusing can be easily evaluated. The figure 6B does not correlate with what I observe in the Movie 1. In this context, I do not see the importance of Figure 7.

Authors should build a good working device first and demonstrate efficient focusing of particles and then test their hypothesis.

Review form: Reviewer 2

Is the manuscript scientifically sound in its present form?

Yes

Are the interpretations and conclusions justified by the results?

Yes

Is the language acceptable?

Yes

Is it clear how to access all supporting data?

Yes

Do you have any ethical concerns with this paper?

No

Have you any concerns about statistical analyses in this paper?

I do not feel qualified to assess the statistics

Recommendation?

Accept as is

Comments to the Author(s)

The authors have well addressed my concerns. I would like to recommend its publication.

Review form: Reviewer 3

Is the manuscript scientifically sound in its present form?

Yes

Are the interpretations and conclusions justified by the results?

Yes

Is the language acceptable?

Yes

Is it clear how to access all supporting data?

Yes

Do you have any ethical concerns with this paper?

No

Have you any concerns about statistical analyses in this paper?

No

Recommendation?

Accept with minor revision (please list in comments)

Comments to the Author(s)

In this paper, the authors developed a glass-based acoustic microfluidic device to investigate the correlation between the degree of acoustic focusing vs. thickness of the microfluidic device. Showing that for the same applied voltage, a higher degree of acoustic focusing of beads was observed in thinner microfluidic devices, and the degree of focussing in a device is positively correlated to the strength of the incoming acoustic waves, thin glass microfluidic devices were found to enhance focussing of submicron particles. These results suggest that acoustic focusing will also be applicable to microorganisms of a few μm in thin glass microfluidic devices.

Overall, the revised manuscript is clearly written and presented. As well, the originality and the contributions are significant to publish the paper in this journal. The reviewer has the following comments/ concerns:

- In the summary refer to the difference of acoustic waves propagating on glass and silicon.
- The conclusions have to be revised and reorganized to be more focused and concise. The conclusions must not just repeat the data. Instead, it should highlight the significant of the finding for the field as well as for general interests.

Decision letter (RSOS-181776.R0)

18-Jan-2019

Dear Dr Tanaka

On behalf of the Editor, I am pleased to inform you that your Manuscript RSOS-181776 entitled "Enhancement in acoustic focusing of micro/nanoparticles by thinning a microfluidic device" has been accepted for publication in Royal Society Open Science subject to minor revision in accordance with the referee suggestions. Please find the referees' comments at the end of this email.

The reviewers and Subject Editor have recommended publication, but also suggest some minor revisions to your manuscript. Therefore, I invite you to respond to the comments and revise your manuscript.

- Ethics statement

- Data accessibility

If you wish to submit your supporting data or code to Dryad (<http://datadryad.org/>), or modify your current submission to dryad, please use the following link:
<http://datadryad.org/submit?journalID=RSOS&manu=RSOS-181776>

- **Competing interests**

- **Authors' contributions**

- **Acknowledgements**

- **Funding statement**

Because the schedule for publication is very tight, it is a condition of publication that you submit the revised version of your manuscript before 27-Jan-2019. Please note that the revision deadline will expire at 00.00am on this date. If you do not think you will be able to meet this date please let me know immediately.

When submitting your revised manuscript, you will be able to respond to the comments made by the referees and upload a file "Response to Referees" in "Section 6 - File Upload". You can use this to document any changes you make to the original manuscript. In order to expedite the

processing of the revised manuscript, please be as specific as possible in your response to the referees.

on behalf of Dr Chong Li (Associate Editor) and R. Kerry Rowe (Subject Editor)
openscience@royalsociety.org

Associate Editor Comments to Author (Dr Chong Li):

Associate Editor

Comments to the Author:

Dear Authors,

Thank you for re-submitting the manuscript after significant efforts on addressing the comments from previous reviewers. However there are still minor concerns with the paper. Please take into account Reviewer 2's comments on focusing capability of the devices and Reviewer 3's comments on "Conclusions" when revising the manuscript.

Yours sincerely

Associate Editor

Dr. Chong Li

Reviewer comments to Author:

Reviewer: 2

Comments to the Author(s)

The authors have well addressed my concerns. I would like to recommend its publication.

Reviewer: 1

Comments to the Author(s)

In this resubmission, authors have tried to address my main comments. Authors have mentioned the reason for using the glass-glass device instead of the more commonly used glass-silicon device; however, I am still not convinced by the performance of this device. Authors have failed to demonstrate good focusing in any of the devices for 10 μm particles even at very slow flow rates (0.2 $\mu\text{L}/\text{min}$). There is so many previous works showing excellent focusing of particles and cells smaller than 10 μm at much higher flow rates than 0.2 $\mu\text{L}/\text{min}$. Since authors have not investigated the most common device type (glass-silicon), to come to a conclusion and use the finding for further studies, authors should at least first establish a working device. Authors have mentioned the reason for using slow flow rate was due to the weak focusing of sub-micron particles. It is not necessary to use sub micron-particles to establish the phenomenon. Acoustic streaming can interfere with particle focusing when the particles are smaller than 1 μm , and the flow rate is very slow. The observation in Movie 1 cannot be considered as good focusing. The end of Movie 1 shows some non-focused particles. The movies are so short that it is difficult to be convinced that the device is capable of good focusing. I would like to see a more extended video clip with unfocused stream getting focused in real time. Also, the video should be collected away from the trifurcate end so that the focusing can be easily evaluated. The figure 6B does not correlate with what I observe in the Movie 1. In this context, I do not see the importance of Figure 7.

Authors should build a good working device first and demonstrate efficient focusing of particles and then test their hypothesis.

Reviewer: 3

Comments to the Author(s)

In this paper, the authors developed a glass-based acoustic microfluidic device to investigate the correlation between the degree of acoustic focusing vs. thickness of the microfluidic device. Showing that for the same applied voltage, a higher degree of acoustic focusing of beads was observed in thinner microfluidic devices, and the degree of focussing in a device is positively correlated to the strength of the incoming acoustic waves, thin glass microfluidic devices were found to enhance focussing of submicron particles. These results suggest that acoustic focusing will also be applicable to microorganisms of a few μm in thin glass microfluidic devices.

Overall, the revised manuscript is clearly written and presented. As well, the originality and the contributions are significant to publish the paper in this journal. The reviewer has the following comments/ concerns:

- In the summary refer to the difference of acoustic waves propagating on glass and silicon.
- The conclusions have to be revised and reorganized to be more focused and concise. The conclusions must not just repeat the data. Instead, it should highlight the significant of the finding for the field as well as for general interests.

Author's Response to Decision Letter for (RSOS-181776.R0)

See Appendix B.

Decision letter (RSOS-181776.R1)

23-Jan-2019

Dear Dr Tanaka,

I am pleased to inform you that your manuscript entitled "Enhancement in acoustic focusing of micro and nanoparticles by thinning a microfluidic device" is now accepted for publication in Royal Society Open Science.

on behalf of Dr Chong Li (Associate Editor) and Professor R. Kerry Rowe (Subject Editor)
openscience@royalsociety.org

Appendix A

Answers to the Reviewers' Comments

Associate Editor Comments to Author (Dr Chong Li):

Thank you for submitting your work to Royal Society Open Science for consideration of publishing. Unfortunately, we regret to reject it as its current form. Based on the feedback from our fellow reviewers, there are major defects in the paper although significant amount of efforts have been spent on investigating the effects of glass thickness on acoustic particle focusing. Please refer to the reviewer's comments for details when considering to resubmit the paper.

Answer: We deeply appreciate all comments given by the editor and reviewers. Below we have highlighted **the given comments in bold**, made our comments as “**Answer**”, and wrote **our modifications on our revised manuscript in red** with the previous description (denoted as “**Before**”) and revised description (denoted as “**After**”) to respond to the given comments/questions/suggestions.

Reviewer: 1

Comments to the Author(s)

Comments:

Authors tried to discuss an essential aspect of acoustophoretic devices which is of clear importance to the acoustofluidic community. However, I am not convinced with the conclusion that the authors made based on the reported data and discussion. It is obvious that the thickness of acoustic devices has an impact on the efficiency of focusing and a study like this should do a thorough investigation without limiting to one type of material as stated below. Overall, the manuscript contains significant weaknesses, therefore, I cannot recommend publishing the manuscript in the current format.

Answer: We highly appreciate the reviewer's comment and would like to answer the reviewer's questions.

In this study, our primary focus was investigation on the correlation between degree of acoustic focusing and thickness of microfluidic devices. We consider that the influence of device thickness on acoustic focusing is not always obvious because acoustic energy can be allocated to both acoustic focusing and deformation of the acoustofluidic device. As the device becomes thinner, the device deformation by acoustic waves is larger. An extreme case of deformation is permanent damage to the device. In fact, our study showed that a 0.2-mm device did not double the degree of acoustic focusing comparing with a 0.4-mm device, and an ultrathin (0.012-mm) glass device was permanently damaged by the acoustic waves.

In this study, we have accumulated the skills and technology to overcome technical difficulties in producing thin glass devices that many research groups have faced. We think this is one major reason why no systematic study has been performed on a correlation between device thickness and degree of acoustic focusing. Thus, we were motivated to study this perspective. Meanwhile, we did not intend to achieve the best acoustic focusing performance or to compare the difference in acoustic focusing among various materials.

To clearly figure out the correlation between device thickness and degree of acoustic focusing, we desired employing a single material to fabricate microfluidics, rather than employing two or more materials as some microfluidic devices are composed of. Thus, we selected glass as the structural material of the microfluidic devices because devices could be made of glass only and glass has optical transparency and can propagate acoustic waves efficiently. Although the obtained results were limited to the glass microfluidic device as the reviewer mentioned, publishing the current work will provide a useful guideline for designing acoustophoretic devices by providing the influence of device thickness on degree of acoustic focusing.

1) Authors used only glass as a substrate to fabricate microfluidic device but, silicon is the most commonly used substrate in bulk acoustic wave-based devices. Glass substrates are rarely used thus I am not convinced about the study just limiting to the glass. Silicon has better acoustic properties than glass. It would make the manuscript convincing if authors compare the Si-devices with different thicknesses and/or different thicknesses of top glass layer which is bonded to Si on such devices. Silicon devices can easily be integrated with optical detection systems by having a glass lid; thus I cannot accept the reason to use glass only devices in this study.

Answer: Thank you for the comment. As the reviewer suggested, silicon is much more frequently employed in acoustofluidics because of its excellent acoustic properties. To find the difference between silicon and glass in propagation of acoustic waves, we conducted additional experiments in which propagated acoustic waves showed 10% higher amplitude on average on silicon than glass. We would like to discuss this result in page 4, lines 35-48 (4.3, 2nd paragraph) and page 5, lines 17-23 (5.1, 1st paragraph).

On the other hand, our primary purpose was to investigate the correlation between the degree of acoustic focusing vs. thickness of the microfluidic device. For such a purpose, it is desirable to use devices composed of a single material to

conduct a clear investigation on how the device thickness influences the degree of acoustic focusing. Thus, we have selected glass as the substrate for the device because it does not require an intermediate or adhesive layer to make a microfluidic device, and it has properties of its optical transparency and ability to conduct acoustic waves efficiently. Although silicon is a great material for propagation of acoustic waves, silicon-based microfluidic devices require two materials, silicon and glass, to perform optical observation. If a silicon-glass device is used, the ratio of silicon and glass affects the correlation between the degree of acoustic focusing and device thickness. To avoid this complexity and by considering that the acoustic properties of glass and silicon were not too much different (no more than 10% difference in vibrational amplitude by acoustic waves), we used devices composed of glass only. As Reviewer 2 admitted, even in this limited condition, this investigation will provide some guidelines for designing glass-based acoustic microfluidics. This discussion on why we decided to work only with glass microfluidic devices is on page 2, lines 24-26 (2. Introduction, 6th paragraph):

Before (Page 4)

(None)

After (Page 4)

To investigate the difference in propagation of acoustic surface waves on glass and silicon, micro-vibrational amplitude of glass induced by acoustic waves was measured based on laser reflection. Both glass and silicon plates had thickness of 0.5 mm. As shown in Fig.S1A, a He-Ne laser (632.8 nm, S145197, Lumentum, Milpitas, CA, USA) emitting a beam diameter of less than 1 mm was irradiated onto a specific point on the substrate. Four points at 15-mm intervals were investigated along the line of the attached piezoelectric transducer. The piezoelectric transducer was glued in place with cyanoacrylate glue and was connected to a function generator for actuation and a home-made power amplifier. The actuation frequency was 3.46 MHz by applying 20 V on the piezoelectric transducer to observe the largest amplitude for both glass and silicon. The reflected

beam was collected by a segmented photodiode (SPOT 2D 1647-1, OSI Optoelectronics, Hawthorne, CA, USA). Fluctuation of the beam induced by micro-vibrations was changed to an electronic signal and amplified by an instrumentation (BB-INA103, Texas Instruments, Dallas, TX, US). The principle of this measurement system resembles that of an atomic force microscope, which is a highly accurate device to measure surface height in a non-contact manner. The difference of collected signals was directly proportional to the difference of the vibrational amplitudes.

Before (Page 5)

(None.)

After (Page 5)

Compared with silicon, a frequently employed material for acoustofluidic devices, glass plate showed 10% smaller amplitude of propagated acoustic waves on average (Fig. S1). Given that the density of silicon is 2.33 g/cm^3 and the density of borosilicate glass is 2.23 g/cm^3 , the difference in the mass of the examined silicon and glass plates was 4.3%. According to equation 1 with these values, energy of the acoustic waves propagating on glass was 22% less than that on silicon. In other words, the 22% energy difference induced by the difference in device thickness corresponded to the material difference of glass and silicon. Therefore, this difference was small compared with the range of glass device thicknesses in this study.

Before (Page 2)

We have selected to use glass for our microfluidics because of its excellent optical transparency beside its efficient propagation of acoustic wave although there are some other materials such as SU-8 to construct a highly sophisticated acoustofluidic device[18].

After (Page 2)

We ~~have~~ selected ~~to use~~ glass ~~to use~~ for our microfluidic devices because of its excellent optical transparency ~~beside~~ and its efficient propagation of acoustic waves although there are ~~some~~ other materials such as silicon and SU-8 that have been used to construct sophisticated acoustofluidic devices with good propagation of acoustic waves [25,26].

2) Page 4 line 48, the authors mentioned the flow rate used is 0.2 $\mu\text{L}/\text{min}$, which is a very low flow rate in common acoustophoresis experiments. Effect of thickness is not described in higher flow rates such as 100 $\mu\text{L}/\text{min}$. The supporting movies indicate that even at this very low flow rate the extent of focusing is poor. Movie 2 clearly shows that particles are not focused and they are directed into three outlets. Previous work from other groups has utilized high flow rates with thick devices yet with very efficient focusing.

Answer: Thank you for the comment. As the reviewer mentioned, the flow rate used in this study was slow although it was sufficient for investigating our primary interest, correlation between degree of acoustic focusing vs. thickness of microfluidic device. In addition, if the flow was as fast as 100 $\mu\text{L}/\text{min}$, our detection system would have faced a difficulty in detecting small particles, especially submicron beads. Thus, the low flow rate was beneficial for our study to evaluate the degree of acoustic focusing of small particles correctly.

In Movie 2, as the reviewer commented, we agree that particles are not focused well because we used a thick microfluidic device (1.8-mm thick) to compare with the results of focusing in a thin device (0.4-mm thick). We expected to have a much lower degree of acoustic focusing in a thick device and the results of thin (Movie 1) and thick (Movie 2) devices matched the expectation. We had an insufficient explanation and would like to modify it on Page 7, lines 25-29 (Section 5.3, 3rd paragraph) as following:

Before

The introduced beads were focused by acoustic radiation through a straight microchannel and collected through the middle channel (Fig. 6c and Movie 1 and 2 of ESM). A portion of the liquid that suspended fluorescent beads flowed into detours as waste. Thus, particles were concentrated by flowing through this microfluidic device when they flowed into the middle channel for collection.

After

The introduced beads were focused by acoustic radiation through a straight microchannel and collected through the middle channel **in the 0.4-mm device (Movie 1) while acoustic focusing was not strong in the 1.8-mm device (Movie 2)**. A portion of the liquid that suspended fluorescent beads flowed into **the** detours as waste. Thus, particles were concentrated by flowing through this microfluidic device when they flowed into the middle channel for collection (**Fig. 6a**).

3) Authors mentioned that they have used 120 μm wide channels. Assuming the device was operated at half wavelength mode, the resonance frequency used (3.98 MHz) does not match with the width (assuming the speed of sound in the medium ~ 1500 m/s).

Answer: Thank you for mentioning this. We had also expected that 3.98 MHz would match well to the channel of approximately 188- μm width for acoustic focusing. However, we did not obtain the results as we expected; inertial force possibly affected acoustic focusing[40]. We empirically found that the channel width of 120 μm worked well for our system. To clarify this, we added the following sentence in page 5, lines 24-26 (5.1, 2nd paragraph):

Before

Fluorescent beads of 2- μm diameter were introduced into a straight microchannel of 120- μm width and 35- μm depth at a flow rate of 0.2 $\mu\text{L}/\text{min}$ to investigate the

correlation between acoustic focusing and the thickness of glass microfluidic devices.

After

Fluorescent beads of 2- μm diameter were introduced into a straight microchannel of 120- μm width and 35- μm depth at a flow rate of 0.2 $\mu\text{L}/\text{min}$ to investigate the correlation between acoustic focusing and the thickness of glass microfluidic devices. **The channel width was empirically adjusted to 120 μm to avoid any disturbance, possibly due to inertial force, on acoustic focusing of particles[40].**

[40] Guo BS et. al. 2018 Optofluidic time-stretch quantitative phase microscopy. *Methods* 136, 116-125.

4) Page 5, Line 51-52: Authors reasoned that they had to use high voltage due to high frequency (3.98 MHz). However, particles should focus at a lower voltage when the resonance frequency is high.

Answer: We appreciate the reviewer for pointing this out. As commented, acoustic radiation force increases as the frequency of acoustic waves is larger. In the manuscript, we tried to state that higher voltage was needed for focusing particles in the combination of higher frequency and narrower channel (vs. the values in reference [21]). Also, the device materials were different in [21] which used silicon and glass. However, our description was confusing and we would like to modify this on page 6, lines 36-38 (5.2 2nd paragraph) as following:

Before

While the present study applies higher voltage to a transducer (20 V vs. 11 V), this difference was caused by different frequencies of acoustic waves (3.98 MHz vs. 3.24 MHz) to induce effective acoustic focusing on the microchannels with the different width (120 μm vs. 230 μm).

After

While the present study applies higher voltage to a transducer (20 V vs. 11 V) with different frequencies of acoustic waves (3.98 MHz vs. 3.24 MHz), this difference could be caused by differences in the width (120 μm vs. 230 μm) and device material (glass vs. glass and silicon) to induce effective acoustic focusing.

5) Design and fabrication of the device are not discussed well in the manuscript.

Answer: Thank you for the comment. We modified section 4.2 (page 4, lines 10-24) to discuss design and fabrication of the device more:

Before

The glass microfluidic device was fabricated by the previously described method[32] with slight modifications. In short, the glass plates were cleaned by $\text{H}_2\text{SO}_4/\text{H}_2\text{O}_2$ solution for 15 minutes to subsequently deposit Cr and Au layers on the glass surface by sputtering (EIS-220, Elionix, Tokyo, Japan). Photoresist (OFPR) was spin-coated on the metal layers and exposed to UV light for 14.5 s to develop the microchannel pattern. Through Cr, Au, and glass etching, the microchannel was patterned onto the glass surface. The etched glass plate and a glass plate with inlet and outlet ports were treated by $\text{H}_2\text{SO}_4/\text{H}_2\text{O}_2$ solution to activate the glass surface. These activated glass plates were firstly bonded manually. For permanent bonding of the glass, a weight of 1.2 kg was placed on top of them and the complex was exposed to a temperature of 620°C for 0.7-mm glass or 500°C for thinner glass. Then, the glass microfluidic device was cut to the dimensions of 70 x 15 (H x W) mm making it more sensitive to propagated acoustic wave.

After

The glass microfluidic device was fabricated by the previously described method[32] with slight modifications. In short, the glass plates were cleaned by

H₂SO₄/H₂O₂ solution for 15 minutes to subsequently deposit Cr and Au layers on the glass surface by sputtering (EIS-220, Elionix, Tokyo, Japan). Photoresist (OFPR) was spin-coated on the metal layers and exposed to UV light for 14.5 s to develop the microchannel pattern. The pattern on the photomask had either a straight microchannel of 6-mm length and 50- μ m width for acoustic focusing experiments or a channel of 5-mm length and 50- μ m width that branched into a middle channel of 0.5-mm length and 1- μ m width and two detours of 1.1-mm length and 140- μ m width for the enrichment of fluorescent particles experiments. Through Cr, Au, and glass etching by corresponding etchant, the microchannel of 35- μ m depth was patterned onto the glass surface. The etched glass plate and a glass plate that had inlet and outlet ports made by a diamond-coated drill of 0.8-mm diameter (DCSSSD0080, Mitsubishi Materials, Tokyo, Japan) were treated with H₂SO₄/H₂O₂ solution to activate the glass surface. These activated glass plates were firstly bonded manually. For permanent bonding of the glass, a weight of 1.2 kg was placed on top of them and the complex was exposed to a temperature of 620°C for 0.7-mm glass or 500°C for thinner glass. Then, the glass microfluidic device was cut to the dimensions of 70 x 15 (H x W) mm making it more sensitive to propagated acoustic waves.

6) Authors claim that thinner glass substrates (<0.2mm) are not useful due to the fragility under acoustic mechanical stress. I am not sure why they still investigated devices with 12 μ m thickness. There is no use of such devices for real applications. (e.g. Pressure drop issues and particle clogging will be significant)

Answer: Related to comment 1), our primary focus was investigation about the degree of acoustic focusing vs. thickness of a microfluidic device. We considered that data of acoustic focusing in an ultrathin device were useful to study more comprehensively if it could be obtained. To state this clearly, we modified the sentence in page 5, lines 48-49 (5.1, 4th paragraph):

Before

In further experiments, we investigated ultrathin glass microfluidic devices of 0.012-mm thickness to check whether an ultrathin device was feasible for acoustic focusing.

After

In further **acoustic focusing** experiments, we investigated ultrathin glass microfluidic devices of 0.012-mm thickness to check whether **it was feasible to use** an ultrathin device was feasible **to find the degree of acoustic focusing in a device with extreme thickness.**

7) The extent of the focusing is calculated based on the distribution of particles inside the microchannels, but no quantitative measurement of the samples collected from each outlet is provided.

Answer: Thank you for the comment and we would like to explain clearly why we chose counting beads inside of the microchannels. The main purpose of sample enrichment in this study was to demonstrate how thickness of the device affects collection of particles via acoustic focusing. Therefore, we decided to count the distribution of particles inside the microchannels, rather than in the collected suspension from each outlet, because the effect of device thickness on acoustic focusing could be investigated more directly by excluding the influence of different dimensions of the middle (collecting) channel and the detour. To explain this, we would like to modify page 7, lines 20-23 (5.3, 3rd paragraph) as the following:

Before

For enrichment of fluorescent polystyrene beads, a piezoelectric transducer was attached to the glass microfluidic channel alongside the straight microchannel. Fluorescent beads were introduced to the microchannel at a flow rate of 0.5 $\mu\text{L}/\text{min}$ for beads of 2 or 6- μm diameter or 1 $\mu\text{L}/\text{min}$ for beads of 10- μm diameter.

After

For enrichment of fluorescent polystyrene beads, a piezoelectric transducer was attached to the glass microfluidic channel alongside the straight microchannel. Fluorescent beads were introduced into the microchannel at a flow rate of 0.5 $\mu\text{L}/\text{min}$ for beads of 2- or 6- μm diameter or 1 $\mu\text{L}/\text{min}$ for beads of 10- μm diameter. The beads were counted where the microchannel branched to the middle channel and the detours to find the effect of device thickness on particle enrichment before being affected by different dimensions of the middle channel and the detour.

8) On page 5 line 17 authors discuss using 280 nm particles for the acoustic focusing experiment. No matter the thickness of the device, it is well accepted fact that manipulation of submicron particles with direct acoustophoresis is really hard. This adds up nothing to the manuscript. And acoustic streaming effects can be significant for nanoparticles at very slow flow rates.

Answer: Thank you for the valuable comment. As the reviewer commented, 280-nm particles are difficult to manipulate while, related to our answers in 1) and 6), this experiment was planned to investigate degree of acoustic focusing in an ultrathin device as a part of systematic study. Although the device was found not to tolerate actuation of the piezoelectric transducer before we could observe the behavior of particles, we hoped to see any results of focusing (focusing via acoustic radiation force, focusing by acoustic streaming effect or no apparent focusing) in the 0.012-mm device to discuss the influence of device thickness on focusing.

9) In usual acoustophoresis setup, 20 Vpp is sufficient to focus the particles >2 μm in size in a very narrow space within the microchannel. Authors do not mention why it is not the case with glass devices.

Answer: When particles with diameter of more than 2 μm were not focused well by applying 20 V to the piezoelectric transducer in this study, we considered that

thickness of glass microfluidics affected the degree of focusing. Since our primary purpose was investigating the degree of acoustic focusing vs. thickness of devices, the influence of material was a different parameter to investigate and beyond the scope of this study. Because of this comment, we modified the following sentence in page 8, lines 25-26 (6. Conclusion, 3rd paragraph):

Before

As theoretically expected, the focusing capability is therefore strongly dependent on the thickness of the device.

After

Although a future investigation is desired to study correlations between acoustic focusing and thickness of microfluidic devices composed of materials other than glass, the acoustic focusing capability is strongly dependent on the thickness of the microfluidic devices composed of glass as theoretically expected.

10) Figures 3b and 5b must have scale bars

Answer: Thank you for the comment. Assuming that the comment mentioned 3b and 5a to add scale bars, we added scale bars in the corresponding figures and explained this in figure captions.

11) The manuscript has some narration inconsistencies and typos.

Answer: Thank you for carefully reading our manuscript. We have checked through the entire manuscript and have corrected English errors as much as possible.

Reviewer: 2

Comments to the Author(s)

The authors considered the effects of glass thickness on acoustic particle focusing. Although the obtained results are limited to glass microfluidic devices, I still think the work is interesting and may provide some guidelines for designing glass-based acoustic microfluidics. In their future works, the authors are strongly encouraged to explore this thickness-effect in devices of other popular materials (e.g., PDMS, SU-8). Overall, I would like to recommend its publication in Royal Society Open Science if the authors are willing to address the following comments.

Answer: We appreciate the reviewer's comment on this study and the suggestion for possible future work. The following are our answers to the reviewer's questions.

(1) I don't know the meaning of "the thickness of a glass microfluidic device" (i.e., page 2, line 29). The authors are suggested to give a clear definition. Does it mean the distance between the transducer and the microchannel or the thickness of the microchannel in the bulk glass?

Answer: We recognize the need to clarify the definition. We meant the thickness of a microfluidic device as total thickness of upper, intermediate (if it is included), and lower plates that compose a microfluidic device. In this study, it was the sum of the thicknesses of the upper glass plate (the plate with inlet and outlet holes) and lower glass plate (the plate with etched microchannels). In other words, it was the distance between the top plane of the upper glass plate and the bottom plane of the lower glass plate. To clarify this point, we added the definition in page 2, lines 14-16 (2. Introduction, 4th paragraph) as the following:

Before

For example, decreasing the thickness of a microfluidic device without changing the microchannel design itself readily increases the amplitude of acoustic waves and avoids the generation of additional heat otherwise caused by increasing the electric input into a piezoelectric transducer.

After

For example, decreasing the thickness (i.e. the distance between the top plane of the upper layer and the bottom plane of the lower layer) of a microfluidic device without changing the microchannel design itself readily increases the amplitude of acoustic waves and avoids the generation of additional heat otherwise caused by increasing the electric input into a piezoelectric transducer.

(2) The simulation of the flow in branched microchannel is very simple and meaningless. It would be better if the authors move it to the supporting information.

Answer: Thank you for the suggestion. We moved Figs. 6a and 6b with their corresponding caption into the supporting information. Thus, the following changes were made:

Before Figs. 6a and 6b -> **After** Figs. S4a and S4b.

Before Figs. 6c and 6d -> **After** Figs. 6a and 6b.

Before Fig. S3 -> **After** Fig. S5.

(3) The focusing performance under the optimal condition is not very good. Is it possible to achieve the single-line focusing using this acoustic focusing device?

Answer: Thank you for the comment. With our setup, 10- μm beads in 0.4-mm device showed focusing close to, but not completely, single-line focusing (Fig. 5b). To achieve single-line focusing, we assume either applying larger amplitude of acoustic waves or employing larger diameter of particles is required. Therefore, we added the following sentence in page 6, lines 27-29 (5.2, 2nd paragraph):

Before

In general, fluorescent beads of larger diameter were focused narrowly in glass microfluidic devices of 0.4 and 2.1-mm thickness, although the beads of each diameter appeared to be less dispersed in a 0.4-mm device as compared to a 2.1-mm device.

After

In general, fluorescent beads of larger diameter were focused narrowly in glass microfluidic devices of 0.4- and 2.1-mm thickness, although the beads of each diameter appeared to be less dispersed in a 0.4-mm device as compared to a 2.1-mm device. **The strongest acoustic focusing was observed for 10- μm beads in the 0.4-mm device while amplitude of acoustic waves or diameter of beads needed to be larger to attain single-line focusing.**

(4) In the introduction, the authors are suggested to briefly discuss the importance of particle focusing (e.g., for particle/cell separation (Analytical chemistry, 2018, 90(6): 4212-4220.) or microflow cytometer (Sensors and Actuators B: Chemical, 2018, 266:26-45; Analytical chemistry, 2017, 89(5): 3154-3161.)). In addition, the other active or passive focusing schemes (e.g., inertial microfluidic, electric, magnetic and optical methods) are suggested to be discussed. This would make the readers better understand the meaning of studying particle focusing and why the acoustic particle focusing is studied in this work.

Answer: We appreciate the suggestion and modified page 1 (2. Introduction, 1st paragraph) like the following with adding the suggested literatures:

Before

Acoustic focusing has gained attention as a non-contact method for manipulating micro/nanoparticles found in biological[1,2] and industrial[3,4] fields. This focusing technique has been applied to sorting[5–7], enrichment[8,9], and analysis[10,11] of particles. For example, acoustic focusing is employed in flow cytometry[12,13], where focused particles need to be precisely aligned in a microchannel to obtain high-quality images in continuous flow.

After

Acoustic focusing has gained attention as a non-contact method for manipulating micro/nanoparticles found in biological[1,2] and industrial[3,4] fields. **Particle focusing, including acoustic focusing, has been widely studied by utilizing inertial force[5] as well as electric[6], magnetic[7] and optical[8] methods. These focusing techniques are highly compatible with quickly advancing technology of miniaturized instruments e.g. microflow cytometry[9,10] that are suitable for manipulating small particles like cells[11].** ~~This~~ Acoustic focusing, as a **particle focusing** technique, has **also** been applied to sorting[12–14], enrichment[15,16], and analysis[17,18] of particles. For example, acoustic focusing is employed in flow cytometry[19,20], where focused particles need to be precisely aligned in a microchannel to obtain high-quality images in continuous flow.

Appendix B

Answers to the Reviewers' Comments

Associate Editor (Dr Chong Li) Comments to Author:

Thank you for re-submitting the manuscript after significant efforts on addressing the comments from previous reviewers. However there are still minor concerns with the paper. Please take into account Reviewer 2's comments on focusing capability of the devices and Reviewer 3's comments on "Conclusions" when revising the manuscript.

Answer: We deeply appreciate all comments given by the editor and reviewers. Below we have highlighted **the given comments in bold**, made our comments as "Answer", and wrote **our modifications on our revised manuscript in red** with the previous description (denoted as "Before") and revised description (denoted as "After") to respond to the given comments/questions/suggestions. In the revised manuscript, we modified Results and discussion (sections 5.3 and 5.4) and Conclusion (section 6) with respect to the Reviewer's comments.

Reviewer: 1

Comments to the Authors:

In this resubmission, authors have tried to address my main comments. Authors have mentioned the reason for using the glass-glass device instead of the more commonly used glass-silicon device; however, I am still not convinced by the performance of this device. Authors have failed to demonstrate good focusing in any of the devices for 10 μm particles even at very slow flow rates (0.2 $\mu\text{L}/\text{min}$). There is so many previous works showing excellent focusing of particles and cells smaller than 10 μm at much higher flow rates than 0.2 $\mu\text{L}/\text{min}$. Since authors have not investigated the most common device type (glass-silicon), to come to a conclusion and use the finding for further studies, authors should at least first establish a working device. Authors have mentioned the reason for using slow flow rate was due to the weak focusing of sub-micron particles. It is not necessary to use sub micron-particles to establish the phenomenon. Acoustic streaming can interfere with particle focusing when the particles are smaller than 1 μm , and the flow rate is very slow. The observation in Movie 1 cannot be considered as good focusing. The end of Movie 1 shows some non-focused particles. The movies are so short that it is difficult to be convinced that the device is capable of good focusing. I would like to see a more extended video clip with unfocused stream getting focused in real time. Also, the video should be collected away from the trifurcate end so that the focusing can be easily evaluated. The figure 6B does not correlate with what I observe in the Movie 1. In this context, I do not see the importance of Figure 7. Authors should build a good working device first and demonstrate efficient focusing of particles and then test their hypothesis.

Answer: We appreciate and would like to answer the reviewer's comments.

This study has focused on investigating the correlation between degree of acoustic focusing and thickness of microfluidic devices while we understand that previous studies have reported excellent focusing of small particles as the reviewer

mentioned. Although we did not investigate acoustic focusing of glass-silicon devices due to absence of glass-silicon bonding equipment in our lab, we compared propagation of acoustic waves on glass and silicon to indicate the potential of glass-silicon devices (section 4.3 and 5.1). Improved acoustic focusing could be accomplished if an amplifier generating higher voltage was available to obtain stronger acoustic waves (Fig. 3) although we do not have this amplifier. Thus, we paid more attention on the varied degree of acoustic focusing in different thickness of the devices rather than the best focusing ability of the devices or the difference of device-composing materials. As the reviewer pointed out, collection of 10- μm beads was not 100% in Movie 1; Fig. 6B, including the result of Movie 1, showed that 10- μm beads were collected at 96.0% on average. Fig. 7 served to state acoustic focusing of microorganisms with confirming previous studies of cellular viability. Since we recognize our focusing results were not optimal (also stated in section 5.2, 2nd paragraph), we would like to modify sections 5.3 and 5.4 based on considering the reviewer's comments:

Section 5.3, 2nd paragraph (page 7)

Before

For enrichment of fluorescent polystyrene beads, a piezoelectric transducer was attached to the glass microfluidic channel alongside the straight microchannel. Fluorescent beads were introduced into the microchannel at a flow rate of 0.5 $\mu\text{L}/\text{min}$ for beads of 2- or 6- μm diameter or 1 $\mu\text{L}/\text{min}$ for beads of 10- μm diameter. The beads were counted where the microchannel branched to the middle channel and the detours to find the effect of device thickness on particle enrichment before being affected by different dimensions of the middle channel and the detour. The introduced beads were focused by acoustic radiation through a straight microchannel and collected through the middle channel in the 0.4-mm device (Movie 1) while acoustic focusing was not strong in the 1.8-mm device (Movie 2). A portion of the liquid that suspended fluorescent beads flowed into the detours as waste. Thus, particles were concentrated by flowing through this microfluidic device when they flowed into the middle channel for collection (Fig. 6a).

After

For enrichment of fluorescent polystyrene beads, a piezoelectric transducer was attached to the glass microfluidic channel alongside the straight microchannel. Fluorescent beads were introduced into the microchannel at a flow rate of 0.5 $\mu\text{L}/\text{min}$ for beads of 2- or 6- μm diameter or 1 $\mu\text{L}/\text{min}$ for beads of 10- μm diameter. The beads were counted where the microchannel branched to the middle channel and the detours to find the effect of device thickness on particle enrichment before being affected by different dimensions of the middle channel and the detour. The introduced beads were focused by acoustic radiation through a straight microchannel and collected through the middle channel in the 0.4-mm device (Movie 1). **Even though it was not optimal focusing to attain collection of all beads in the 0.4-mm device, while** acoustic focusing was **much stronger in the 0.4-mm device than ~~not strong~~** in the 1.8-mm device (Movie 2). A portion of the liquid that suspended fluorescent beads flowed into the detours as waste. Thus, particles were concentrated by flowing through this microfluidic device when they flowed into the middle channel for collection (Fig. 6a).

Section 5.4, 3rd paragraph (page 8)

Before

These results suggest that acoustic focusing will also be applicable to microorganisms of a few μm in thin glass microfluidic devices, even though this field still requires further scientific development. As current technologies employ acoustic focusing for enhancing screening and analysis of microorganisms in flow systems such as flow cytometry, thin glass microfluidic devices offer new possibilities to further enhance the efficiency of acoustic-focusing systems.

After

These results suggest that acoustic focusing will also be applicable to microorganisms of a few μm in thin glass microfluidic devices, even though **our device was not optimized for this focusing and** this field still requires further

scientific development. As current technologies employ acoustic focusing for enhancing screening and analysis of microorganisms in flow systems such as flow cytometry, thin glass microfluidic devices offer new possibilities to further enhance the efficiency of acoustic-focusing systems.

Reviewer: 2

Comments to the Authors:

The authors have well addressed my concerns. I would like to recommend its publication.

Answer: We appreciate the reviewer's comment and the decision.

Reviewer: 3

Comments to the Authors:

In this paper, the authors developed a glass-based acoustic microfluidic device to investigate the correlation between the degree of acoustic focusing vs. thickness of the microfluidic device. Showing that for the same applied voltage, a higher degree of acoustic focusing of beads was observed in thinner microfluidic devices, and the degree of focussing in a device is positively correlated to the strength of the incoming acoustic waves, thin glass microfluidic devices were found to enhance focussing of submicron particles. These results suggest that acoustic focusing will also be applicable to microorganisms of a few μm in thin glass microfluidic devices.

Overall, the revised manuscript is clearly written and presented. As well, the originality and the contributions are significant to publish the paper in this journal. The reviewer has the following comments/ concerns:

- In the summary refer to the difference of acoustic waves propagating on glass and silicon.**
- The conclusions have to be revised and reorganized to be more focused and concise. The conclusions must not just repeat the data. Instead, it should highlight the significant of the finding for the field as well as for general interests.**

Answer: We highly appreciate the reviewer's comment. We would like to modify the Conclusion based on the comments.

Section 6 (page 8)

Before

In this study, we used glass acoustofluidic devices to investigate how acoustic focusing of flowing particles was influenced by the thickness of the employed glass microfluidic devices. We show that the thickness of glass microfluidic devices has a

strong influence on the focusing of small particles by acoustic radiation. Furthermore, we show that thin glass microfluidic devices are able to focus submicron particles in a straight path via acoustic radiation. We also demonstrated that acoustic focusing was applicable to concentrating particles by removal of excess liquid, and to acoustic focusing of *E. coli* of one to a few μm in size.

Using thin glass microfluidic devices, it is possible to adopt more diverse layouts of microchannels that may not be advantageous in terms of the energy of acoustic waves but are desirable in special cases e.g. when using shallow microchannels for observation of particles in fast flow. Another application that thin glass microfluidic devices can contribute to is the isolation and analysis of rare cells and organelles via acoustic focusing. Because this application requires manipulation of small particles and concentration of particles to trap them in small volumes, employing a combination of acoustic focusing and thin glass microfluidic devices is an obvious method to achieve this. Separating mixed particles with diverse sizes is another possible application. Particles of different sizes can be sorted based on their size-dependent dispersion properties in acoustic focusing.

Overall, we showed that thin glass microfluidic devices significantly enhance acoustic focusing of particles, including focusing of submicron particles by acoustic radiation force, due to their highly efficient energy propagation of acoustic waves to target particles. Although a future investigation is desired to study correlations between acoustic focusing and thickness of microfluidic devices composed of materials other than glass, the acoustic focusing capability is strongly dependent on the thickness of the microfluidic devices composed of glass as theoretically expected. Engineering the device thickness as a method to enhance acoustic focusing does not require a change of equipment or microchannel dimensions, effectively eliminating the two major practical difficulties that hamper high acoustic focusing efficiencies in current setups. Finally, we expect thin glass microfluidic devices to become an essential component in future microfluidic systems that require optimized acoustic focusing.

After

In this study, we used glass acoustofluidic devices to investigate how acoustic focusing of flowing particles was influenced by the thickness of the employed glass microfluidic devices. ~~We show that the thickness of glass microfluidic devices has a strong influence on the~~ Thin glass microfluidic devices significantly enhanced focusing of small particles by acoustic radiation force, due to their highly efficient energy propagation of acoustic waves to target particles. Although acoustic waves propagated 10% smaller amplitude on glass than silicon, glass was suitable for our investigation because glass microfluidic devices could be produced without intermediate layer, adhesive materials, or specialized apparatus. ~~Furthermore, we show that thin glass microfluidic devices are able to focus submicron particles in a straight path via acoustic radiation. We also~~ In this setup, we demonstrated that acoustic focusing was applicable to concentrating particles by removal of excess liquid, and to acoustic focusing of *E. coli* of one to a few μm in size.

Using thin glass microfluidic devices, it is possible to adopt more diverse layouts of microchannels that may not be advantageous in terms of the energy propagation of acoustic waves but are desirable in special cases e.g. when using shallow microchannels for observation of particles in fast flow[44]. ~~Another Possible applications that thin glass microfluidic devices can contribute to is~~ are the isolation and analysis of rare cells and organelles with imaging[44-46], as well as separating mixed particles with diverse sizes[47,48]. ~~via acoustic focusing. Because this application requires manipulation of small particles and concentration of particles to trap them in small volumes, employing a combination of acoustic focusing and thin glass microfluidic devices is an obvious method to achieve this. Separating mixed particles with diverse sizes is another possible application. Particles of different sizes can be sorted based on their size dependent dispersion properties in acoustic focusing.~~

~~Overall, we showed that thin glass microfluidic devices significantly enhance acoustic focusing of particles, including focusing of submicron particles by acoustic radiation force, due to their highly efficient energy propagation of acoustic waves to target particles.~~ Although a future investigation is desired to study correlations

between acoustic focusing and thickness of microfluidic devices composed of materials other than glass, the acoustic focusing capability is strongly dependent on the thickness of the microfluidic devices composed of glass as theoretically expected. Engineering the device thickness as a method to enhance acoustic focusing does not require a change of equipment or microchannel dimensions, effectively eliminating the two major practical difficulties that hamper high acoustic focusing efficiencies in current setups. Finally, we expect thin glass microfluidic devices to become an essential component in future microfluidic systems that require optimized acoustic focusing.